# CHANNEL-WISE INFLUENCE: ESTIMATING DATA INFLUENCE FOR MULTIVARIATE TIME SERIES

## ABSTRACT

The influence function, a robust statistics technique, is an effective post-hoc method that measures the impact of modifying or removing training data on model parameters, offering valuable insights into model interpretability without requiring costly retraining. It would provide extensions like increasing model performance, improving model generalization, and offering interpretability. Recently, **M**ultivariate **T**ime **S**eries (MTS) analysis has become an important yet challenging task, attracting significant attention. However, there is no preceding research on the influence functions of MTS to shed light on the effects of modifying the channel of MTS. Given that each channel in an MTS plays a crucial role in its analysis, it is essential to characterize the influence of different channels. To fill this gap, we propose a channel-wise influence function, which is the first method that can estimate the influence of different channels in MTS, utilizing a first-order gradient approximation. Additionally, we demonstrate how this influence function can be used to estimate the influence of a channel in MTS. Finally, we validated the accuracy and effectiveness of our influence estimation function in critical MTS analysis tasks, such as MTS anomaly detection and MTS forecasting. According to abundant experiments on real-world datasets, the original influence function performs worse than our method and even fails for the channel pruning problem, which demonstrates the superiority and necessity of the channel-wise influence function in MTS analysis.

## 1 INTRODUCTION

Multivariate time series (MTS) plays an important role in a wide variety of domains, including internet services (Dai et al., 2021) , industrial devices (Finn et al., 2016; Oh et al., 2015) , health care (Choi et al., 2016b;a), finance (Maeda et al., 2019; Gu et al., 2020) , and so on. Thus, MTS modeling is crucial across a wide array of applications, including disease forecasting, traffic forecasting, anomaly detection, and action recognition. In recent years, researchers have focused on deep learning-based MTS analysis methods (Zhou et al., 2021; Tuli et al., 2022; Xu et al., 2023; Liu et al., 2024; Xu et al., 2021; Wu et al., 2022). Due to the large number of different channels in MTS, numerous studies aim to analyze the importance of these channels (Liu et al., 2024; Zhang & Yan, 2022; Nie et al., 2022; Wang et al., 2024). Some of them concentrate on using graph or attention structure to capture the channel dependencies (Liu et al., 2024; Deng & Hooi, 2021), while some of them try to use Channel Independence to enhance the generalization ability on different channels of time series model (Nie et al., 2022; Zeng et al., 2023). Although these deep learning methods have achieved state-of-the-art performance, most of these methods focus on understanding the MTS by refining the model architecture to improve their models' performance.

Different from previous work, we try to better understand MTS from a data-centric perspective-influence function (Hampel, 1974; Koh & Liang, 2017). The influence function is proposed to study the counterfactual effect between training data and model performance. For independent and identically distributed (i.i.d.) data, influence functions estimate the model's change when there is an infinitesimal perturbation added to the training distribution, e.g., a reweighing on some training instances and dataset pruning, which has been widely used in computer vision and natural language processing tasks, achieving promising results (Yang et al., 2023; Tan et al., 2024; Thakkar et al., 2023; Cohen et al., 2020; Chen et al., 2020; Pruthi et al., 2020). Considering that, it is essential to develop an appropriate influence function for MTS. It would provide extensions like increasing

model performance, improving model generalization, and offering interpretability of the interactions between the channels and the time series models.

To the best of our knowledge, the influence of MTS in deep learning has not been studied, and it is nontrivial to apply the original influence function in Koh & Liang (2017) to this scenario. Since different channels of MTS usually include different kinds of information and have various relationships (Wu et al., 2020; Liu et al., 2024), the original influence function can not distinguish the influence of different channels in MTS because it is designed for a whole data sample, according to the definition of the original influence function. In addition, our experiments also demonstrate that the original influence function does not support anomaly detection effectively and fails to solve the forecasting generalization problem in MTS, while it performs well on computer vision and natural language process tasks (Yang et al., 2023; Thakkar et al., 2023). Thus, how to estimate the influence of different channels in MTS is a critical problem. Considering a well-designed influence function should be able to distinguish the influence between different channels, we propose a channel-wise influence function, a first-order gradient approximation (Pruthi et al., 2020), to characterize the influence of different channels. Then, we introduce how to use this function in MTS anomaly detection (Saquib Sarfraz et al., 2024) and MTS forecasting (Liu et al., 2024) tasks effectively. Finally, we use various kinds of experiments on real-world datasets to demonstrate the characteristics of our novel influence function and prove it can be widely used in the MTS analysis tasks.

The main contributions of our work are summarized as follows:

- We developed a novel channel-wise influence function, a first-order gradient approximation, which is the first of its kind to effectively estimate the channel-wise influence of MTS.
- We designed two channel-wise influence function-based algorithms for MTS anomaly detection and MTS forecasting tasks, and validated its superiority and necessity.
- We discovered that the original functions do not perform well on MTS anomaly detection tasks and cannot solve the forecasting generalization problem.
- Experiments on various real-world datasets illustrate the superiority of our method on the MTS anomaly detection and forecasting tasks compared with original influence function. Specifically, our influence-based methods rank top-1 among all methods for comparison.

## 2 RELATED WORK

### 2.1 BACKGROUND OF INFLUENCE FUNCTIONS

Influence functions estimate the effect of a given training example, $z'$, on a test example, $z$, for a pre-trained model. Specifically, the influence function approximates the change in loss for a given test example z when a given training example $z'$ is removed from the training data and the model is retrained. Koh & Liang (2017) derive the aforementioned influence to be $I(z', z) := \nabla_{\boldsymbol{\theta}} L(z'; \theta)^\top \boldsymbol{H}_{\boldsymbol{\theta}}^{-1} \nabla_{\boldsymbol{\theta}} L(z; \boldsymbol{\theta})$, where $\boldsymbol{H}_{\boldsymbol{\theta}}$ is the loss Hessian for the pre-trained model: $\boldsymbol{H}_{\boldsymbol{\theta}} := 1/n \sum_{i=1}^{n} \nabla_{\boldsymbol{\theta}}^2 L(z; \boldsymbol{\theta})$, evaluated at the pre-trained model's final parameter checkpoint. The loss Hessian is typically estimated with a random mini-batch of data. The main challenge in computing influence is that it is impractical to explicitly form $\boldsymbol{H}_{\boldsymbol{\theta}}$ unless the model is small, or if one only considers parameters in a few layers. TracIn (Pruthi et al., 2020) address this problem by utilizing a first-order gradient approximation: $\text{TracIn}(z', z) := \nabla_{\boldsymbol{\theta}} L(z'; \boldsymbol{\theta})^\top \nabla_{\boldsymbol{\theta}} L(z; \boldsymbol{\theta})$, which has been proved effectively in various tasks (Thakkar et al., 2023; Yang et al., 2023; Tan et al., 2024).

### 2.2 BACKGROUND OF MULTIVARIATE TIME SERIES

There are various types of MTS analysis tasks. In this paper, we mainly focus on unsupervised anomaly detection and preliminarily explore the value of our method in MTS forecasting.

**MTS Anomaly detection:** MTS anomaly detection has been extensively studied, including complex deep learning models (Su et al., 2021; Tuli et al., 2022; Deng & Hooi, 2021; Xu et al., 2022). These models are trained to forecast or reconstruct presumed normal system states and then deployed to detect anomalies in unseen test datasets. The anomaly score, defined as the magnitude of prediction or reconstruction errors, serves as an indicator of abnormality at each timestamp. However,

Saquib Sarfraz et al. (2024) have demonstrated that these methods create an illusion of progress due to flaws in the datasets (Wu & Keogh, 2021) and evaluation metrics (Kim et al., 2022), and they provide a more fair and reliable benchmark.

**MTS Forecasting:** In MTS forecasting, many methods try to model the temporal dynamics and channel dependencies effectively. An important issue in MTS forecasting is how to better generalize to unseen channels with a limited number of channels (Liu et al., 2024). This places high demands on the model architecture, as the model must capture representative information across different channels and effectively utilize this information. There are two popular state-of-the-art methods to achieve this. One is iTransformer (Liu et al., 2024), which uses attention mechanisms to capture channel correlations. The other is PatchTST (Nie et al., 2022), which enhances the model's generalization ability by sharing the same model parameters across different channels through a Channel-Independence strategy. However, both of these methods are model-centric methods, which cannot identify the most informative channels in the training data for the model.

Although MTS forecasting and anomaly detection are two different kinds of tasks, both of their state-of-the-art methods have fully utilized the channel information in the MTS through model-centric methods. Different from previous model-centric methods, we propose a data-centric method to improve the model's performance on MTS downstream tasks and identify practical techniques to improve the analysis of the training data by leveraging channel-wise information.

## 3 CHANNEL-WISE INFLUENCE FUNCTION

The influence function (Koh & Liang, 2017) requires the inversion of a Hessian matrix, which is quadratic in the number of model parameters. Additionally, the representer point method necessitates a complex, memory-intensive line search or the use of a second-order solver such as LBFGS. Fortunately, the original influence function can be accelerated and approximated by TracIn (Pruthi et al., 2020) effectively. TracIn is inspired by the fundamental theorem of calculus. The fundamental theorem of calculus decomposes the difference between a function at two points using the gradients along the path between the two points. Analogously, TracIn decomposes the difference between the loss of the test point at the end of training versus at the beginning of training along the path taken by the training process. The specific definition can be derived as follows:

$$\text{TracIn}\left(\boldsymbol{z}', \boldsymbol{z}\right) = L\left(\boldsymbol{z}; \boldsymbol{\theta}\right) - L(\boldsymbol{z}; \boldsymbol{\theta}') \approx \eta \nabla_{\boldsymbol{\theta}} L\left(\boldsymbol{z}'; \boldsymbol{\theta}\right)^{\top} \nabla_{\boldsymbol{\theta}} L(\boldsymbol{z}; \boldsymbol{\theta}) \tag{1}$$

where $\boldsymbol{z}'$ is the training example, $\boldsymbol{z}$ is the testing example, $\boldsymbol{\theta}$ is the model parameter, $\boldsymbol{\theta}'$ is the updated parameter after training with $\boldsymbol{z}'$, $L(\cdot)$ is the loss function and $\eta$ is the learning rate during the training process, which defines the influence of training $\boldsymbol{z}'$ on $\boldsymbol{z}$.

However, in the MTS analysis, the data sample $\boldsymbol{z}$, $\boldsymbol{z}'$ are MTS, which means TracIn can only calculate the whole influence of all channels. In other words, it fails to characterize the difference between different channels. To fill this gap we derive a new channel-wise influence function, using a derivation method similar to TracIn. Thus, we obtain Theorem 3.1 which formulates the channel-wise influence matrix, and the proof can be found in Appendix 6.

**Theorem 3.1. (Channel-wise Influence function)** *Assuming the $\boldsymbol{c}'_i$, $\boldsymbol{c}_j$ is the i-th channel and j-th channel from the data sample $\boldsymbol{z}'$, $\boldsymbol{z}$ respectively, $\boldsymbol{\theta}$ is the well-trained parameter of the model, $L(\cdot)$ is the loss function and $\eta$ is the learning rate during the training process. The first-order approximation of the original influence function can be derived at the channel-wise level as follows:*

$$TracIn\left(\boldsymbol{z}', \boldsymbol{z}\right) = \sum_{i=1}^{N} \sum_{j=1}^{N} \eta \nabla_{\boldsymbol{\theta}} L\left(\boldsymbol{c}'_i; \boldsymbol{\theta}\right)^{\top} \cdot \nabla_{\boldsymbol{\theta}} L\left(\boldsymbol{c}_j; \boldsymbol{\theta}\right) \tag{2}$$

Given the result, we define a channel-wise influence matrix $\boldsymbol{M}_{CInf}$ and each element $a_{i,j}$ in it can be described as $a_{i,j} := \eta \nabla_{\boldsymbol{\theta}} L\left(\boldsymbol{c}'_i; \boldsymbol{\theta}\right)^{\top} \cdot \nabla_{\boldsymbol{\theta}} L\left(\boldsymbol{c}_j; \boldsymbol{\theta}\right)$. Thus, according to the theorem 3.1, the original TracIn can be treated as a sum of these elements in the channel-wise influence matrix $\boldsymbol{M}_{CInf}$, failing to utilize the channel-wise information in the matrix specifically. Considering that, the final channel-wise influence function can be defined as follows:

$$\text{CIF}\left(\boldsymbol{c}'_i, \boldsymbol{c}_j\right) := \eta \nabla_{\boldsymbol{\theta}} L\left(\boldsymbol{c}'_i; \boldsymbol{\theta}\right)^{\top} \cdot \nabla_{\boldsymbol{\theta}} L\left(\boldsymbol{c}_j; \boldsymbol{\theta}\right) \tag{3}$$

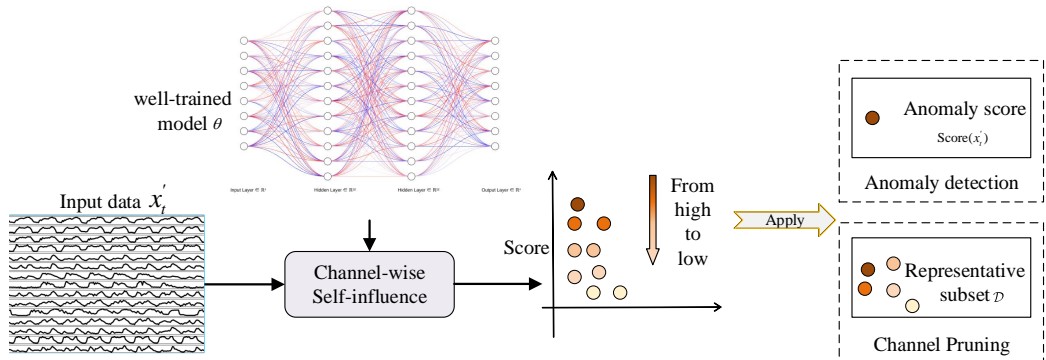

Figure 1: The framework of applying channel-wise influence function. The specific calculation method for the Score is detailed in Algorithms 1 and 2.

where $c_i, c'_j$ is the i-th channel and j-th channel from the data sample $z, z'$ respectively, $\theta$ is the well-trained parameter of the model and $\eta$ is the learning rate during the training process. This channel-wise influence function describes the influence between different channels among MTS.

**Remark 3.2. (Characteristics of Channel-wise Influence Matrix)** *The channel-wise influence matrix reflects the relationships between different channels in a specific model. Specifically, each element $a_{i,j}$ in the matrix $M_{CInf}$ represents how much training with channel i helps reduce the loss for channel j, which means similar channels usually have high influence score. Each model has its unique channel influence matrix, reflecting the model's way of utilizing channel information in MTS. Therefore, we can use $M_{CInf}$ for post-hoc interpretable analysis of the model.*

## 4 APPLICATION IN MTS ANALYSIS

In this section, we focus on two important tasks in MTS analysis: MTS anomaly detection and MTS forecasting. We discuss the relationship between our channel-wise influence function and these tasks, and then explain how to apply our method to these critical problems.

### 4.1 MTS ANOMALY DETECTION

**Problem Definition:** Defining the training MTS as $\boldsymbol{x} = \{\boldsymbol{x}_1, \boldsymbol{x}_2, ..., \boldsymbol{x}_T\}$, where $T$ is the duration of $\boldsymbol{x}$ and the observation at time $t$, $\boldsymbol{x}_t \in \mathbb{R}^N$, is a $N$ dimensional vector where $N$ denotes the number of channels, thus $\boldsymbol{x} \in \mathbb{R}^{T \times N}$. The training data only contains non-anomalous timestep. The test set, $\boldsymbol{x}' = \{\boldsymbol{x}'_1, \boldsymbol{x}'_2, ..., \boldsymbol{x}'_T\}$ contains both normal and anomalous timestamps and $\boldsymbol{y}' = [\boldsymbol{y}'_1, \boldsymbol{y}'_2, ..., \boldsymbol{y}'_T] \in \{0, 1\}$ represents their labels, where $\boldsymbol{y}'_t = 0$ denotes a normal and $\boldsymbol{y}'_t = 1$ an anomalous timestamp t. Then the task of anomaly detection is to select a function $f_{\boldsymbol{\theta}} : X \to R$ such that $f_{\boldsymbol{\theta}}(\boldsymbol{x}_t) = \boldsymbol{y}_t$ estimates the anomaly score. When it is larger than the threshold, the data is predicted anomaly.

**Relationship between self-influence and anomaly score:** According to the conclusion in Section 4.1 of (Pruthi et al., 2020), influence can be an effective way to detect the anomaly sample. Specifically, the idea is to measure self-influence, i.e., the influence of a training point on its own loss, i.e., the training point $\boldsymbol{z}'$ and the test point $\boldsymbol{z}$ in TracIn are identical. From an intuitive perspective, self-influence reflects how much a model can reduce the loss during testing by training on sample $\boldsymbol{z}'$ itself. Therefore, anomalous samples, due to their distribution being inconsistent with normal training data, tend to reduce more loss, resulting in a greater self-influence. Therefore, when we sort test examples by decreasing self-influence, an effective influence computation method would tend to rank anomaly samples at the beginning of the ranking.

**Apply in MTS anomaly detection:** Based on these premises, we propose to derive an anomaly score based on the channel-wise influence function 3 for MTS. Consider a test sample $\boldsymbol{x}'$ for which we wish to assess whether it is an anomaly. We can compute the channel-wise influence matrix $M_{CInf}$ at first and then get the diagonal elements of the $M_{CInf}$ to indicate the anomaly score of each channel. Since, according to the Remark 3.2 and the nature of self-influence, the diagonal elements reflect the channel-wise self-influence, it is an effective method to reflect the anomaly level of each channel. Consistent with previous anomaly detection methods, we use the maximum anomaly

---

**Algorithm 1** Channel-wise influence based MTS anomaly detection

---

**Require:** test dataset $\mathcal{D}_{test}$; a well-trained network $\boldsymbol{\theta}$; loss function $L(\cdot)$; threshold $h$
    empty anomaly score dictionary $\rightarrow$ ADscore[]; empty prediction dictionary $\rightarrow$ ADPredict[]
    **for** $\boldsymbol{x} \in \mathcal{D}_{test}$ **do**
        $ADscore\,[\boldsymbol{x}] = \underset{i}{max}(\eta \nabla_{\boldsymbol{\theta}} L\,(\boldsymbol{c}'_i; \boldsymbol{\theta})^{\top} \cdot \nabla_{\boldsymbol{\theta}} L\,(\boldsymbol{c}'_i; \boldsymbol{\theta}))$
    **end for**
    Normalize $ADScore[\cdot]$ ;                     /* Anomaly score normalization. */
    **if** $ADscore\,[\boldsymbol{x}] > h$ **then**
        $ADPredict\,[\boldsymbol{x}] = 1$ ;                          /* Anomaly sample. */
    **else**
        $ADPredict\,[\boldsymbol{x}] = 0$ ;                          /* Normal sample. */
    **end if**
    return anomaly detection result $ADPredict\,[\cdot]$.

---

score across different channels as the anomaly score of MTS $\boldsymbol{x}'$ at time $t$ as:

$$\text{Score}\,(\boldsymbol{x}'_t) := \underset{i}{max}(\eta \nabla_{\boldsymbol{\theta}} L\,(\boldsymbol{c}'_i; \boldsymbol{\theta})^{\top} \cdot \nabla_{\boldsymbol{\theta}} L\,(\boldsymbol{c}'_i; \boldsymbol{\theta})) \tag{4}$$

where $\boldsymbol{c}'_i$ is the i-th channel of the MTS sample $\boldsymbol{x}'_t$, $\boldsymbol{\theta}$ is the trained parameter of the model, and $\eta$ is the learning rate during the training process. To ensure a fair comparison, we adopt the same anomaly score normalization and threshold selection strategy as outlined in Saquib Sarfraz et al. (2024) for detecting anomalies. Details regarding this methodology can be found in Appendix B. The comprehensive process for MTS anomaly detection is further elaborated in Algorithm 1 and Fig. 1.

## 4.2 MTS FORECASTING

**Forecasting Generalization Problem Definition:** Defining the MTS as $\boldsymbol{x} = \{\boldsymbol{x}_1, \boldsymbol{x}_2, ..., \boldsymbol{x}_T\}$, where $T$ is the duration of $\boldsymbol{x}$ and the observation at time $t$, $\boldsymbol{x}_t \in \mathbb{R}^{N'}$, is a $N'$ dimensional vector where $N'$ denotes the number of channels used in the training process, thus $\boldsymbol{x} \in \mathbb{R}^{T \times N'}$. The aim of multivariate time series forecasting generalization is to predict the future value of $\boldsymbol{x}_{T+1:T+T',n}$, where $T'$ is the number of time steps in the future and the observation at time $t'$, $\boldsymbol{x}_{t'} \in \mathbb{R}^N$, is a $N$ dimensional vector where $N$ is the number of whole channels which is large than $N'$.

**Motivation:** Considering the excellent performance of the influence function in dataset pruning tasks (Tan et al., 2024; Yang et al., 2023) and the generalization issues faced in MTS forecasting mentioned in Section 2, we propose a new task suitable for MTS to validate the effectiveness of our channel-wise influence function named channel pruning. With the help of channel pruning, we can accurately identify the subset of channels that are most representative for the model's training without retraining the model, resulting in helping the model better generalize to unknown channels with a limited number of channels. The definition of the task is described in the following paragraph.

**Channel Pruning Problem Definition:** Given an MTS $\boldsymbol{x} = \{\boldsymbol{c}_1, ..., \boldsymbol{c}_N\}$, $\boldsymbol{y} = \{\boldsymbol{c}'_1, ..., \boldsymbol{c}'_N\}$ containing N channels where $\boldsymbol{c}_i \in R^T$, $\boldsymbol{x}$ is the input space and $\boldsymbol{y}$ is the label space. The goal of channel pruning is to identify a set of representative channel samples from $\boldsymbol{x}$ as few as possible to reduce the training cost and find the relationship between model and channels. The identified representative subset, $\hat{\mathcal{D}} = \{\hat{\boldsymbol{c}}_1, ..., \hat{\boldsymbol{c}}_m\}$ and $\hat{\mathcal{D}} \subset \mathcal{D}$, should have a maximal impact on the learned model, i.e. the test performances of the models learned on the training sets before and after pruning should be very close, as described below:

$$\mathbb{E}_{\boldsymbol{c} \sim P(\mathcal{D})} L(\boldsymbol{c}, \theta) \simeq \mathbb{E}_{\boldsymbol{c} \sim P(\mathcal{D})} L\,(\boldsymbol{c}, \theta_{\hat{\mathcal{D}}}) \tag{5}$$

where $P(\mathcal{D})$ is the data distribution, $L(\cdot)$ is the loss function, and $\boldsymbol{\theta}$ and $\boldsymbol{\theta}_{\hat{\mathcal{D}}}$ are the empirical risk minimizers on the training set $\mathcal{D}$ before and after pruning $\hat{\mathcal{D}}$, respectively, i.e., $\boldsymbol{\theta} = \arg\min_{\boldsymbol{\theta} \in \Theta} \frac{1}{n} \sum_{\boldsymbol{c}_i \in \mathcal{D}} L\,(\boldsymbol{c}_i, \boldsymbol{\theta})$ and $\boldsymbol{\theta}_{\hat{\mathcal{D}}} = \arg\min_{\boldsymbol{\theta} \in \Theta} \frac{1}{m} \sum_{\boldsymbol{c}_i \in \hat{\mathcal{D}}} L\,(\boldsymbol{c}_i, \boldsymbol{\theta})$.

**Apply in channel pruning:** Considering the channel pruning problem, our proposed channel-wise self-influence method can effectively address this issue. According to the Remark 3.2, our approach can use $M_{CInf}$ to represent the characteristics of each channel by calculating the influence of different channels. Then, We use a concise approach to obtain a representative subset of channels.

---

**Algorithm 2** Channel-wise influence based MTS channel pruning

---

**Require:** val dataset $\mathcal{D}_{val}$; a well-trained network $\boldsymbol{\theta}$; loss function $L(\cdot)$; sample interval $t$

    empty channel set $\hat{\mathcal{D}} \rightarrow \{\}$ ; empty channel score dictionary $\rightarrow$ CScore[]

    **for** $\boldsymbol{x} \in \mathcal{D}_{val}$ **do**

      **for** $\boldsymbol{c}_i \in \boldsymbol{x}$ **do**

        $CScore\,[\boldsymbol{c}_i] + = \eta \nabla_{\boldsymbol{\theta}} L\left(\boldsymbol{c}_i; \boldsymbol{\theta}\right)^{\top} \cdot \nabla_{\boldsymbol{\theta}} L\left(\boldsymbol{c}_i; \boldsymbol{\theta}\right)$

      **end for**

    **end for**

    Sort(CScore) ;            `/* Sort the influence scores in ascending order. */`

    i=0

    **while** $i < N$ **do**

      **if** $i == t$ **then**

        add $\boldsymbol{c}_i$ to $\hat{\mathcal{D}}$ ;             `/* Sample at regular intervals. */`

      **end if**

      $i+ = 1$

    **end while**

    return pruned channel set $\hat{\mathcal{D}}$.

---

Specifically, we can rank the diagonal elements of $M_{CInf}$, i.e., the channel-wise self-influence, and select the subset of channels at regular intervals for a certain model. Since similar channels have a similar self-influence, we can adopt regular sampling on the original channel set $\mathcal{D}$ based on the channel-wise self-influence to acquire a representative subset of channels $\hat{\mathcal{D}}$ for a certain model and dataset, which is typically much smaller than the original dataset. The detailed process of channel pruning is shown in Algorithm 2 and Fig.1. Consequently, we can train or fine-tune the model with a limited set of data efficiently. Additionally, it can serve as an explainable method to reflect the channel-modeling ability of different approaches. Specifically, the smaller the size of the representative subset $\hat{\mathcal{D}}$ for a method, the fewer channels' information it uses for predictions, and vice versa. In other words, a good MTS modeling method should have a large size of $\hat{\mathcal{D}}$.

## 5 EXPERIMENTS

In this section, we mainly discuss the performance of our method in MTS anomaly detection and explore the value and feasibility of our method in MTS forecasting tasks. All the datasets used in our experiments are real-world and open-source MTS datasets.

### 5.1 MUTIVARIATE TIME SERIES ANOMALY DETECTION

#### 5.1.1 BASELINES AND EXPERIMENTAL SETTINGS

We conduct model comparisons across five widely-used anomaly detection datasets: SMD(Su et al., 2019), MSL (Hundman et al., 2018), SMAP (Hundman et al., 2018), SWaT (Mathur & Tippenhauer, 2016), and WADI (Deng & Hooi, 2021), encompassing applications in service monitoring, space/earth exploration, and water treatment. Since SMD,

Table 1: The detailed dataset information.

| Dataset | Sensors(traces) | Train | Test | Anomalies |
|---------|-----------------|-------|------|-----------|
| SWaT | 51 | 47520 | 44991 | 4589(12.2%) |
| WADI | 127 | 118750 | 17280 | 1633(9.45%) |
| SMD | 38(28) | 25300 | 25300 | 1050(4.21%) |
| SMAP | 25(54) | 2555 | 8070 | 1034(12.42%) |
| MSL | 55(27) | 2159 | 2730 | 286(11.97%) |

SMAP, and MSL datasets contain traces with various lengths in both the training and test sets, we report the average length of traces and the average number of anomalies among all traces per dataset. The detailed information of the datasets can be found in Table. 1.

Given the point-adjustment evaluation metric is proved not reasonable (Saquib Sarfraz et al., 2024; Kim et al., 2022), we use the standard precision, recall and F1 score to measure the performance, which aligns with (Saquib Sarfraz et al., 2024). Moreover, due to the flaws in the previous methods, Saquib Sarfraz et al. (2024) provide a more fair benchmark, including many simple but effective methods, such as GCN-LSTM, PCA ERROR and so on, labeled as **Simple baseline** in the Table 2. Thus, for a fair comparison, we follow the same data preprocessing procedures as described in

Saquib Sarfraz et al. (2024) and use the results cited from their paper or reproduced with their code as strong baselines. Considering iTransformer (Liu et al., 2024) is an effective time series model that can capture the channel dependencies with attention block adaptively, we also add iTransformer as a new baseline. The summary of training details is provided in Appendix B.

Table 2: Experimental results for SWaT, SMD, MSL, SMAP, and WADI datasets. The bold and underlined marks are the best and second-best value. F1: the standard F1 score; P: Precision; R: Recall. For all metrics, higher values indicate better performance.

| Method | Datasets | | | | | | | | | | | | | | |
|---|---|---|---|---|---|---|---|---|---|---|---|---|---|---|---|
| | SWAT | | | SMD | | | SMAP | | | MSL | | | WADI | | |
| | F1 | P | R | F1 | P | R | F1 | P | R | F1 | P | R | F1 | P | R |
| DAGMM (Zong et al., 2018) | 77.0 | 99.1 | 63.0 | 43.5 | 56.4 | 49.7 | 33.3 | 39.5 | 56.0 | 38.4 | 40.1 | 59.6 | 27.9 | 99.3 | 16.2 |
| OmniAnomaly (Su et al., 2019) | 77.3 | 99.0 | 63.4 | 41.5 | 56.6 | 46.4 | 35.1 | 37.2 | 62.5 | 38.7 | 40.7 | 61.5 | 28.1 | 100 | 16.3 |
| USAD (Audibert et al., 2020) | 77.2 | 98.8 | 63.4 | 42.6 | 54.6 | 47.4 | 31.9 | 36.5 | 40.2 | 38.6 | 40.2 | 61.1 | 27.9 | 99.3 | 16.2 |
| GDN Deng & Hooi (2021) | 81.0 | 98.7 | 68.6 | 52.6 | 59.7 | 56.5 | 42.9 | 48.2 | 63.1 | 44.2 | 38.6 | 62.4 | 34.7 | 64.3 | 23.7 |
| TranAD (Tuli et al., 2022) | 80.0 | 99.0 | 67.1 | 45.7 | 57.9 | 48.1 | 35.8 | 37.8 | 52.5 | 38.1 | 40.1 | 59.7 | 34.0 | 29.3 | 40.4 |
| AnomalyTransformer (Xu et al., 2022) | 76.5 | 94.3 | 64.3 | 42.6 | 41.9 | 52.8 | 31.1 | 42.3 | 60.4 | 33.8 | 31.3 | 59.8 | 20.9 | 12.2 | 74.3 |
| PCA ERROR (Simple baseline) | 83.3 | 96.5 | 73.3 | 57.2 | 61.1 | 58.4 | 39.2 | 43.4 | 65.5 | 42.6 | 39.6 | 63.5 | 50.1 | 88.4 | 35.0 |
| 1-Layer MLP (Simple baseline) | 77.1 | 98.1 | 63.5 | 51.4 | 59.8 | 57.4 | 32.3 | 43.2 | 58.7 | 37.3 | 34.2 | 64.8 | 26.7 | 83.4 | 15.9 |
| Single block MLPMixer (Simple baseline) | 78.0 | 85.4 | 71.8 | 51.2 | 60.8 | 55.4 | 36.3 | 45.1 | 61.2 | 39.7 | 34.1 | 62.8 | 27.5 | 86.2 | 16.3 |
| Single Transformer block (Simple baseline) | 78.7 | 86.8 | 72.0 | 48.9 | 58.9 | 53.6 | 36.6 | 42.4 | 62.9 | 40.2 | 42.7 | 56.9 | 28.9 | 90.8 | 17.2 |
| 1-Layer GCN-LSTM (Simple baseline) | 82.9 | 98.2 | 71.8 | 55.0 | 62.7 | 59.9 | 42.6 | 46.9 | 61.6 | 46.3 | 45.6 | 58.2 | 43.9 | 74.4 | 31.1 |
| Using Channel-wise Influence (Ours) | 82.9 | 98.0 | 71.8 | 58.8 | 63.5 | 62.2 | 48.0 | 54.3 | 59.6 | 47.1 | 41.1 | 67.6 | 47.2 | 54.5 | 41.6 |
| Inverted Transformer (Liu et al., 2024) | 83.7 | 96.3 | 74.1 | 55.9 | 65.0 | 57.0 | 39.6 | 49.7 | 60.8 | 45.5 | 44.8 | 66.6 | 48.8 | 64.2 | 39.4 |
| Using Channel-wise Influence (Ours) | 84.0 | 96.4 | 74.4 | 59.1 | 63.6 | 63.8 | 46.3 | 52.9 | 61.3 | 46.1 | 41.9 | 68.4 | 50.5 | 58.7 | 44.2 |

### 5.1.2 MAIN RESULTS

In this experiment, we compare our channel-wise self-influence method with other model-centric methods. Apparently, Table 2 showcases the superior performance of our method, achieving the highest F1 score among the previous state-of-the-art (SOTA) methods. The above results demonstrate the effectiveness of our channel-wise influence function and channel-wise self-influence-based anomaly detection method. Specifically, the use of model gradient information in self-influence highlights that the gradient information across different layers of the model enables the identification of anomalous information, contributing to good performance in anomaly detection.

### 5.1.3 ADDITIONAL ANALYSIS

In this section, we conduct several experiments to validate the effectiveness of the channel-wise influence function and explore the characteristics of the channel-wise influence function.

**Ablation Study:** In our method, the most important part is the design of channel-wise influence and replacing the reconstructed or predicted error with our channel-wise self-influence to detect the anomalies. We conduct ablation studies on different datasets and models. Fig 2a and Fig 2b show that the channel-wise influence is better than the original influence function and the original influence function is worse than the reconstructed error. It is because that the original influence function fails to distinguish which channel is abnormal more specifically. Additionally, both figures demonstrate that our method achieves strong performance across different model architectures, underscoring the effectiveness and generalization capability of our data-centric approach. Given the superiority of our channel-wise influence function over the original influence function, the design of a dedicated channel-wise influence function becomes essential.

**Generalization Analysis:** To demonstrate the generalizability of our method, we applied our channel-wise influence function to various model architectures and presented the results in the following table 3. As clearly shown in the table, our method consistently exhibited superior performance across different model architectures. Therefore, we can conclude that our method is suitable for different types of models, proving that it is a qualified data-centric approach. The full results of the generalization analysis can be found in Table. 7 in the Appendix.

**Parameter Analysis:** According to the formula Eq. 3, we need to compute the model's gradient. Considering computational efficiency, we use the gradients of a subset of the model's parameters to calculate influence. Therefore, we tested the relationship between the number of param-

Table 3: The generalization ability of our method is evaluated in combination with different model architectures on various datasets. Bold marks indicate the best results.

| | Method | 1-Layer MLP | | | Single block MLPMixer | | | Single Transformer block | | |
|---|---|---|---|---|---|---|---|---|---|---|
| | Dataset | F1 | P | R | F1 | P | R | F1 | P | R |
| SMD | Reconstruct Error | 51.4 | 59.8 | 57.4 | 51.2 | 60.8 | 55.4 | 48.9 | 58.9 | 53.6 |
| | Channel-wise Influence | **55.9** | 63.1 | 60.6 | **55.5** | 64.8 | 58.3 | **52.1** | 62.9 | 58.2 |
| SMAP | Reconstruct Error | 32.3 | 43.2 | 58.7 | 36.3 | 45.1 | 61.2 | 36.6 | 42.4 | 62.9 |
| | Channel-wise Influence | **47.0** | 54.5 | 60.9 | **48.0** | 57.5 | 58.9 | **48.5** | 54.1 | 64.6 |
| MSL | Reconstruct Error | 37.3 | 34.2 | 64.8 | 39.7 | 34.1 | 62.8 | 40.2 | 42.7 | 56.9 |
| | Channel-wise Influence | **45.8** | 42.2 | 65.4 | **46.2** | 44.6 | 57.1 | **47.7** | 42.8 | 64.9 |

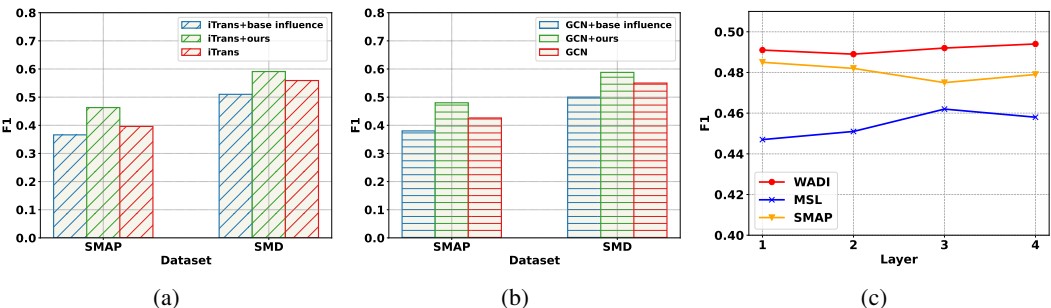

(a)           (b)           (c)

Figure 2: (a)-(b): The ablation study of channel-wise influence function for iTransformer and GCN-LSTM on SMAP and SMD dataset. (c): The relationship between the number of parameters used to calculate gradients and the anomaly detection performance on different datasets.

eters used and the anomaly detection performance, with the results shown in Fig. 2c. Specifically, we use the GCN-LSTM model as an example. The GCN-LSTM model has an MLP decoder, which contains two linear layers, each with weight and bias parameters. Therefore, we can identify four layers of parameters to calculate the gradient and use these four parameters to test the effect of the number of parameters used. The results in Fig. 2c indicate that our method is not sensitive to the choice of parameters. Hence, using only the gradients of the last layer of the network is sufficient to achieve excellent performance in approximating the influence.

**Visualization of Anomaly Score:** To highlight the differences between our channel-wise self-influence method and traditional reconstruction-based methods, we visualized the anomaly scores obtained from the SMAP dataset. Apparently, as indicated by the red box in Fig. 3, the reconstruction error fails to fully capture the anomalies, making it difficult to distinguish some normal samples from the anomalies, as their anomaly scores are similar to the threshold. The results show that our method can detect true anomalies more accurately compared to reconstruction-based methods, demonstrating the advantage of channel-wise influence.

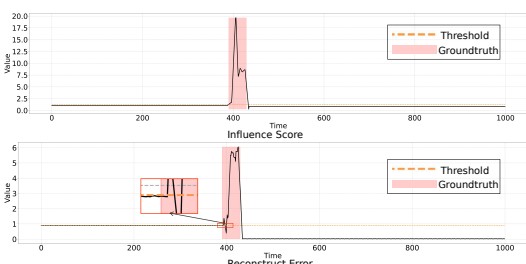

Figure 3: Visual illustration of the anomaly score of different methods.

## 5.2 MULTIVARIATE TIME SERIES FORECASTING

### 5.2.1 CHANNEL PRUNING EXPERIMENT

**Set Up:** To demonstrate the effectiveness of our method, we designed a channel pruning experiment. In this experiment, we selected three datasets with a large number of channels for testing: Electricity

with 321 channels, Solar-Energy with 137 channels, and Traffic with 821 channels. The detailed information of these datasets can be found in the Table 4.

According to Eq.5, the specific aim of the experiment was to determine how to retain only $N\%$ of the channels while maximizing the model's generalization ability across all channels. In addition to our proposed method, we compared it with some naive baseline methods, including training with the first $N\%$ of the channels and randomly selecting $N\%$ of the channels for training. $N$ is changed to demonstrate the channel-pruning ability of these methods.

Table 4: The detailed dataset information.

| Dataset | Dim | Prediction Length | Datasize | Frequency |
|---|---|---|---|---|
| Electricity | 321 | 96 | (18317, 2633, 5261) | Hourly |
| Solar-Energy | 137 | 96 | (36601, 5161, 10417) | 10min |
| Traffic | 862 | 96 | (12185, 1757, 3509) | Hourly |

Table 5: Variate generalization experimental results for Electricity, Solar Energy, and Traffic datasets. We use the MSE metric to reflect the performance of different methods. The bold marks are the best. The predicted length is 96. The red markers indicate the proportion of channels that need to be retained to achieve the original prediction performance.

| Dataset | | ECL | | | | | Solar | | | | | Traffic | | | | |
|---|---|---|---|---|---|---|---|---|---|---|---|---|---|---|---|---|
| Proportion of variables retained | | 5% | 10% | 15% | 20% | 50% | 5% | 10% | 15% | 20% | 50% | 5% | 10% | 15% | 20% | 30% |
| iTransformer | Continuous selection | 0.208 | 0.188 | 0.181 | 0.178 | 0.176 | 0.241 | 0.228 | 0.225 | 0.224 | 0.215 | 0.470 | 0.437 | 0.409 | 0.406 | 0.404 |
| | Random selection | 0.205 | 0.182 | 0.177 | 0.175 | 0.165 | 0.240 | 0.229 | 0.225 | 0.223 | 0.217 | 0.450 | 0.415 | 0.404 | 0.404 | 0.403 |
| | Influence selection | **0.187** | **0.174** | **0.170** | **0.165** | **0.150** | **0.229** | **0.224** | **0.220** | **0.219** | **0.210** | **0.419** | **0.405** | **0.398** | **0.397** | **0.395** |
| | Full variates | | 0.148 | | | | | 0.206 | | | | | 0.395 | | | |
| Proportion of variables retained | | 5% | 10% | 15% | 20% | 45% | 5% | 10% | 15% | 20% | 20% | 5% | 10% | 15% | 20% | 20% |
| PatchTST | Continuous selection | 0.304 | 0.222 | 0.206 | 0.202 | 0.203 | 0.250 | 0.244 | 0.240 | 0.230 | 0.230 | 0.501 | 0.478 | 0.474 | 0.476 | 0.476 |
| | Random selection | 0.230 | 0.208 | 0.202 | 0.196 | 0.186 | 0.242 | 0.240 | 0.235 | 0.230 | 0.230 | 0.495 | 0.478 | 0.467 | 0.464 | 0.464 |
| | Influence selection | **0.205** | **0.191** | **0.190** | **0.186** | **0.176** | **0.228** | **0.226** | **0.226** | **0.223** | **0.223** | **0.483** | **0.470** | **0.456** | **0.452** | **0.452** |
| | Full variates | | 0.176 | | | | | 0.224 | | | | | 0.454 | | | |

**Results Analysis:** The bold mark results in the Table.5 indicate that, when retaining the same proportion of channels, our method significantly outperforms the other two methods. Besides, the red mark results in the table also show that our method can maintain the original prediction performance while using no more than half of the channels, significantly outperforming other baseline methods. These results prove the effectiveness of our method in selecting the representative subsets of channels. Considering our selection strategy is different from conventional wisdom, such as selecting the most influence samples, we add new experiments in Appendix C.1. The results prove that the conventional way to utilize channel-wise influence function cannot work well in channel pruning problem.

In addition to the superior performance shown in the table, our experiment highlights a certain relationship between the model and the channels. Specifically, since iTransformer (Liu et al., 2024) needs to capture channel correlations, it requires a higher retention ratio to achieve the original prediction performance. In contrast, PatchTST (Nie et al., 2022) employs a Channel-Independence strategy, meaning all channels share the same parameters, and therefore, fewer variables are needed to achieve the original prediction performance. This also explains why its predictive performance is not as good as that of iTransformer, as it does not fully learn information from more channels.

**Outlook:** Based on the above results, we believe that in addition to using the channel-wise influence function for channel pruning to improve the efficiency of model training and fine-tuning, another important application is its use as a post-hoc interpretable method to evaluate a model's quality. As our experimental results demonstrate, a good model should be able to fully utilize the information between different channels. Therefore, to achieve the original performance, such a method would require retaining a higher proportion of channels.

### 5.2.2 COMPARING DATA PRUNING WITH CHANNEL PRUNING

**Set up:** To further demonstrate the superiority of channel pruning, we conducted a comparative experiment between data pruning and channel pruning. Specifically, we reduced the data using two pruning strategies: for data pruning, we applied MoSo (Tan et al., 2024), an effective data pruning approach, alongside random data pruning, which involved randomly selecting data samples for pruning. For channel pruning, we utilized our channel-wise influence function. In this experiment, we compared each pruning method at the same remaining ratio. For example, when the horizontal

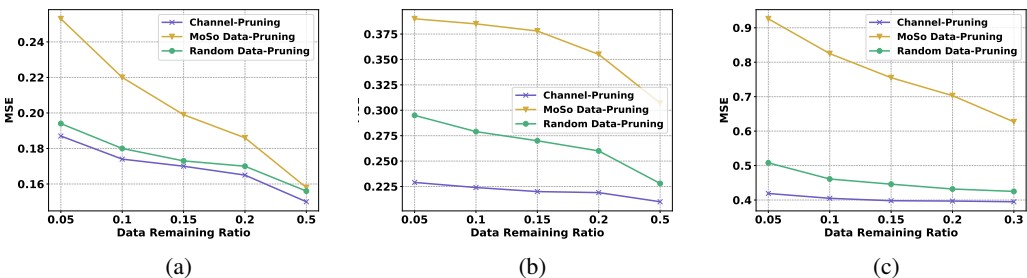

(a)           (b)           (c)

Figure 4: (a)-(c): The comparison experiment between data pruning and channel pruning on different datasets. From left to right are the Electricity dataset, the Solar Energy dataset, and the Traffic dataset. The evaluation metric used is mean squared error (MSE), with lower values indicating better performance. The horizontal axis means the remaining ratio of the dataset.

axis in Fig. 4 indicates that $50\%$ is retained, it means the size of the entire dataset is reduced to half of its original size. In the case of data pruning, half of the training samples will be discarded; whereas in channel pruning, half of the channels will be discarded.

**Result Analysis:** As shown in Fig. 4, our channel pruning method achieved better performance while retaining the same proportion of data on all settings. This suggests that channel pruning is a more suitable method for reducing MTS data than data pruning. Additionally, we previously highlighted the value of channel pruning as a post-hoc method for analyzing MTS models. Therefore, we believe that channel pruning holds greater exploratory value in MTS tasks.

Furthermore, we found that the performance of the MoSo-based pruning method was not as effective as that of the random pruning method. We believe this may be due to the traditional influence method underlying MoSo, which assumes that each data sample is calculated in isolation. However, the samples in time series forecasting usually have strong temporal dependencies, thus resulting in the failure of the MoSo method. Therefore, we consider designing an effective data pruning method specifically for time series forecasting to be a noteworthy open problem.

## 6 CONCLUSIONS

In this paper, we propose a novel influence function that is the first influence function that can estimate the influence of each channel in MTS, which is a concise data-centric method, distinguishing it from previously proposed model-centric methods. In addition, according to abundant experiments on real-world datasets, the original influence function performs worse than our method in anomaly detection and cannot solve the channel pruning problem. This limitation arises from its inability to differentiate the influence across various channels. In contrast, our channel-wise influence function serves as a more universal and effective tool for addressing a wide range of MTS analysis tasks. In conclusion, we believe that our method has significant potential for application and can serve as an effective post-hoc approach for MTS analysis, helping us to better understand the characteristics of MTS and helping us develop more effective MTS models.

**Limitation:** While we have successfully applied our method to two fundamental MTS tasks and demonstrated its effectiveness, there remains a vast landscape of MTS-related tasks that are yet to be explored and understood. Looking ahead, a primary focus of our research will be the further application of the channel-wise influence function. We believe that delving deeper into this area will yield valuable insights and contribute significantly to advancing the field.

**Broader Impact:** Our model is well-suited for multivariate time series analysis tasks, offering practical and positive impacts across various domains, including disease forecasting, traffic prediction, internet services, content delivery networks, wearable devices, and action recognition. However, we emphatically discourage its application in activities related to financial crimes or any other endeavors that could lead to negative societal consequences.

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

# A  PROOF OF THEOREM

*Proof.* The proof of channel-wise influence function:

$$
\begin{aligned}
\text{TracIn}\left(\boldsymbol{z}', \boldsymbol{z}\right) &= L\left(\boldsymbol{z}; \boldsymbol{\theta}\right) - L(\boldsymbol{z}; \boldsymbol{\theta}') \\
&= \sum_{i=1}^{N} L\left(\boldsymbol{c}_i; \boldsymbol{\theta}\right) - \sum_{j=1}^{N} L(\boldsymbol{c}_j; \boldsymbol{\theta}') \\
&= \sum_{i=1}^{N} \left(\nabla L\left(\boldsymbol{c}_i; \boldsymbol{\theta}\right) \cdot \left(\boldsymbol{\theta}' - \boldsymbol{\theta}\right) + O\left(\left\|\boldsymbol{\theta}' - \boldsymbol{\theta}\right\|^2\right)\right) \\
&\approx \sum_{i=1}^{N} \nabla L\left(\boldsymbol{c}_i; \boldsymbol{\theta}\right) \cdot \eta \nabla L\left(\boldsymbol{z}'; \boldsymbol{\theta}\right) \\
&= \sum_{i=1}^{N} \sum_{j=1}^{N} \eta \nabla L\left(\boldsymbol{c}_i; \boldsymbol{\theta}\right) \cdot \nabla L\left(\boldsymbol{c}'_j; \boldsymbol{\theta}\right)
\end{aligned}
\tag{6}
$$

where the first equation is the original definition of TracIn; we rectify the equation and derive the second equation, indicating the sum of the loss of each channel. The third equation is calculated by the first approximation of the loss function and then we replace $(\boldsymbol{\theta}' - \boldsymbol{\theta})$ with $\eta \nabla L\left(\boldsymbol{z}'; \boldsymbol{\theta}\right)$. Therefore, we can derive the final equation which demonstrates the original Influence function at the channel-wise level.

The proof is complete.

$\square$

# B  DETAILS OF EXPERIMENTS

## B.1  TRAINING DETAILS

All experiments were implemented using PyTorch and conducted on a single NVIDIA GeForce RTX 3090 24GB GPU.

**For anomaly detection:** Models were trained using the SGD optimizer with Mean Squared Error (MSE) loss. For both of them, when trained in reconstructing mode, we used a time window of size 10.

**For channel pruning:** Models were trained using the Adam optimizer with Mean Squared Error (MSE) loss. The input length is 96 and the predicted length is 96.

## B.2  ANOMALY SCORE NORMALIZATION

Anomaly detection methods for multivariate datasets often employ normalization and smoothing techniques to address abrupt changes in prediction scores that are not accurately predicted. In this paper, we mainly use two normalization methods, mean-standard deviation and median-IQR, which aligns with Saquib Sarfraz et al. (2024). The details are as follows:

$$
s_i = \frac{\mathrm{S}_i - \widetilde{\mu}_i}{\tilde{\sigma}_i}
\tag{7}
$$

**For median-IQR:** The $\widetilde{\mu}$ and $\tilde{\sigma}$ are the median and inter-quartile range (IQR2) across time ticks of the anomaly score values respectively.

**For mean-standard deviation:** The $\widetilde{\mu}$ and $\tilde{\sigma}$ are the mean and standard across time ticks of the anomaly score values respectively.

For a fair comparison, we select the best results of the two normalization methods as the final result, which aligns with Saquib Sarfraz et al. (2024).

## B.3 THRESHOLD SELECTION

Typically, the threshold which yields the best F1 score on the training or validation data is selected. This selection strategy aligns with Saquib Sarfraz et al. (2024), for a fair comparison.

# C ADDITIONAL MODEL ANALYSIS

## C.1 UTILIZATION OF CHANNEL-WISE INFLUENCE

We conducted new experiments comparing different selecting strategy based on channel-wise influence. The results, shown in the table, indicate that our equidistant sampling approach is more effective than selecting the most influence samples. This is because it covers a broader range of channels, allowing the model to learn more general time-series patterns during training.

Table 6: Variate generalization experimental results for Electricity, Solar Energy, and Traffic datasets. We use the MSE metric to reflect the performance of different methods. The bold marks are the best. The predicted length is 96. The red markers indicate the proportion of channels that need to be retained to achieve the original prediction performance.

| Dataset | | ECL | | | | | Solar | | | | | Traffic | | | | |
|---|---|---|---|---|---|---|---|---|---|---|---|---|---|---|---|---|---|
| Proportion of variables retained | | 5% | 10% | 15% | 20% | 50% | 5% | 10% | 15% | 20% | 50% | 5% | 10% | 15% | 20% | 30% |
| iTransformer | Most influence sample | 0.360 | 0.224 | 0.181 | 0.176 | 0.160 | 0.351 | 0.241 | 0.237 | 0.236 | 0.220 | 0.461 | 0.421 | 0.407 | 0.401 | 0.399 |
| | Ours | **0.187** | **0.174** | **0.170** | **0.165** | **0.150** | **0.229** | **0.224** | **0.220** | **0.219** | **0.210** | **0.419** | **0.405** | **0.398** | **0.397** | **0.395** |
| | Full variates | | 0.148 | | | | | 0.206 | | | | | 0.395 | | | |

## C.2 GENERALIZATION RESULTS

To demonstrate the generalizability of our method, we applied our channel-wise influence function to various model architectures and presented the results in the following table 7. As clearly shown in the table, our method consistently exhibited superior performance across different model architectures. Therefore, we can conclude that our method is suitable for different types of models, proving that it is a qualified data-centric approach.

Table 7: Full results of the generalization ability experiment.

| Method | | 1-Layer MLP | | | Single block MLPMixer | | | Single Transformer block | | |
|---|---|---|---|---|---|---|---|---|---|---|
| Dataset | | F1 | P | R | F1 | P | R | F1 | P | R |
| SMD | Reconstruct Error | 51.4 | 59.8 | 57.4 | 51.2 | 60.8 | 55.4 | 48.9 | 58.9 | 53.6 |
| | Channel-wise Influence | **55.9** | 63.1 | 60.6 | **55.5** | 64.8 | 58.3 | **52.1** | 62.9 | 58.2 |
| SMAP | Reconstruct Error | 32.3 | 43.2 | 58.7 | 36.3 | 45.1 | 61.2 | 36.6 | 42.4 | 62.9 |
| | Channel-wise Influence | **47.0** | 54.5 | 60.9 | **48.0** | 57.5 | 58.9 | **48.5** | 54.1 | 64.6 |
| MSL | Reconstruct Error | 37.3 | 34.2 | 64.8 | 39.7 | 34.1 | 62.8 | 40.2 | 42.7 | 56.9 |
| | Channel-wise Influence | **45.8** | 42.2 | 65.4 | **46.2** | 44.6 | 57.1 | **47.7** | 42.8 | 64.9 |
| SWAT | Reconstruct Error | 77.1 | 98.1 | 63.5 | 78.0 | 85.4 | 71.8 | 78.7 | 86.8 | 72.0 |
| | Channel-wise Influence | **80.1** | 87.7 | 73.7 | **80.6** | 97.6 | 68.6 | **81.9** | 97.7 | 70.6 |
| WADI | Reconstruct Error | 26.7 | 83.4 | 15.9 | 27.5 | 86.2 | 16.3 | 28.9 | 90.8 | 17.2 |
| | Channel-wise Influence | **44.3** | 84.6 | 30.0 | **46.6** | 83.0 | 32.4 | **47.5** | 71.3 | 35.6 |

## C.3 ADDITIONAL DATASET AND BASELINE RESULTS

To demonstrate the effectiveness of our approach, we validated our channel-pruning method on new datasets. Additionally, we incorporated a new baseline, DLinear, a time series forecasting method based on a channel-independence strategy. The specific results are shown below:

**New dataset analysis:**

Since the original number of channels in ETTh1 and ETTm1 is only 7, the horizontal axis in the table directly represents the number of retained channels.

Table 8: The additional dataset results of the channel-pruning experiment.

| Dataset | | ETTh1 | | | ETTm1 | | |
|---|---|---|---|---|---|---|---|
| number of channels retained | | 7 | 3 | 2 | 7 | 3 | 2 |
| iTransformer | Continuous selection | 0.396 | 0.502 | 0.573 | 0.332 | 0.756 | 0.826 |
| | Random selection | 0.396 | 0.428 | 0.434 | 0.332 | 0.362 | 0.372 |
| | Influence selection | 0.396 | 0.403 | 0.420 | 0.332 | 0.333 | 0.355 |
| PatchTST | Continuous selection | 0.400 | 0.460 | 0.491 | 0.330 | 0.539 | 0.687 |
| | Random selection | 0.400 | 0.415 | 0.424 | 0.330 | 0.352 | 0.364 |
| | Influence selection | 0.400 | 0.400 | 0.405 | 0.330 | 0.336 | 0.347 |

The results in the table demonstrate the effectiveness of channel pruning based on the channel-wise influence function, highlighting that PatchTST and iTransformer exhibit comparable utilization of channel information on the ETTh1 and ETTm1 datasets.

**New forecasting length analysis:**

We have added experimental results for the prediction length of 192. The detailed results are as follows:

Table 9: The 192 forecasting length of the channel-pruning experiment.

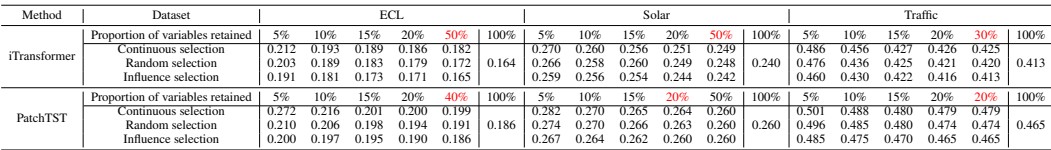

| Method | Dataset | ECL | | | | | | Solar | | | | | | Traffic | | | | | |
|---|---|---|---|---|---|---|---|---|---|---|---|---|---|---|---|---|---|---|---|
| | Proportion of variables retained | 5% | 10% | 15% | 20% | 50% | 100% | 5% | 10% | 15% | 20% | 50% | 100% | 5% | 10% | 15% | 20% | 30% | 100% |
| iTransformer | Continuous selection | 0.212 | 0.193 | 0.189 | 0.186 | 0.182 | | 0.270 | 0.260 | 0.256 | 0.251 | 0.249 | | 0.486 | 0.456 | 0.427 | 0.426 | 0.425 | |
| | Random selection | 0.203 | 0.189 | 0.183 | 0.179 | 0.172 | 0.164 | 0.266 | 0.258 | 0.260 | 0.249 | 0.248 | 0.240 | 0.476 | 0.436 | 0.425 | 0.421 | 0.420 | 0.413 |
| | Influence selection | 0.191 | 0.181 | 0.173 | 0.171 | 0.165 | | 0.259 | 0.256 | 0.254 | 0.244 | 0.242 | | 0.460 | 0.430 | 0.422 | 0.416 | 0.413 | |
| PatchTST | Proportion of variables retained | 5% | 10% | 15% | 20% | 40% | 100% | 5% | 10% | 15% | 20% | 50% | 100% | 5% | 10% | 15% | 20% | 20% | 100% |
| | Continuous selection | 0.272 | 0.216 | 0.201 | 0.200 | 0.199 | | 0.282 | 0.270 | 0.265 | 0.264 | 0.260 | | 0.501 | 0.488 | 0.480 | 0.479 | 0.479 | |
| | Random selection | 0.210 | 0.206 | 0.198 | 0.194 | 0.191 | 0.186 | 0.274 | 0.270 | 0.266 | 0.263 | 0.260 | 0.260 | 0.496 | 0.485 | 0.480 | 0.474 | 0.474 | 0.465 |
| | Influence selection | 0.200 | 0.197 | 0.195 | 0.190 | 0.186 | | 0.267 | 0.264 | 0.262 | 0.260 | 0.260 | | 0.485 | 0.475 | 0.470 | 0.465 | 0.465 | |

From the results shown in the table, it can be observed that channel-pruning based on channel-wise influence is more effective. Additionally, iTransformer still exhibits a larger core subset, demonstrating its superior ability to model channel dependency.

**New baseline analysis:**

Table 10: The channel-pruning experiment results of DLinear model.

| Dataset | | ECL | | | | | | Solar | | | | | | Traffic | | | | | |
|---|---|---|---|---|---|---|---|---|---|---|---|---|---|---|---|---|---|---|---|
| Proportion of variables retained | | 5% | 10% | 15% | 20% | 50% | 100% | 5% | 10% | 15% | 20% | 50% | 100% | 5% | 10% | 15% | 20% | 30% | 100% |
| DLinear | Continuous selection | 0.201 | 0.200 | 0.198 | 0.197 | 0.196 | | 0.311 | 0.309 | 0.307 | 0.301 | 0.301 | | 0.649 | 0.647 | 0.645 | 0.645 | 0.645 | |
| | Random selection | 0.200 | 0.198 | 0.196 | 0.196 | 0.196 | 0.196 | 0.306 | 0.304 | 0.303 | 0.301 | 0.301 | 0.301 | 0.649 | 0.648 | 0.645 | 0.645 | 0.645 | 0.645 |
| | Influence selection | 0.197 | 0.196 | 0.196 | 0.196 | 0.196 | | 0.301 | 0.301 | 0.301 | 0.301 | 0.301 | | 0.646 | 0.645 | 0.645 | 0.645 | 0.645 | |

The experimental results in the table show that the core channel subset of DLinear is less than 5%, which highlights the limited ability of simple linear models to utilize information from different channels effectively.

## C.4 Additional Complexity analysis results

To illustrate the complexity of our method, we added complexity analysis experiments in both time series anomaly detection and forecasting tasks. In these experiments, we measured the time required to compute the influence of all channels of a single multivariate time series data sample.

**Anomaly detection:**

We have added an experiment measuring the time required for detection at each time point to demonstrate the complexity of our approach, as shown in the table below:

Table 11: The time required for our method on different time series model.

| Dataset | GCN_lstm+ours | iTransformer+ours |
|---------|---------------|-------------------|
| SWAT | 1.4ms | 1.5ms |
| WADI | 6.4ms | 6.5ms |

The results in the table indicate that our detection speed is at the millisecond level, which is acceptable for real-world scenarios.

**Channel-pruning:**

By measuring the time required for calculating single-instance influence, we demonstrated how the computational time scales with the number of channels.

Table 12: The time required for channel-pruning method on different time series datasets.

| | ETTm1 | Solar-Energy | Electricity | traffic |
|---|-------|--------------|-------------|---------|
| iTransformer+ours | 0.0025s | 0.023s | 0.071s | 0.18s |

From the table, it can be observed that the computational complexity approximately increases linearly with the number of channels.

## C.5 COMPARING WITH OTHER CHANNEL PRUNING METHOD

To better highlight the effectiveness of our method, we compared it with the approach proposed in the paper(Gu et al., 2021), referred to as NFS. The specific results are as follows:

Table 13: The comparing of different channel-pruning methods.

| Dataset | | ECL | | | | | | Solar | | | | | | Traffic | | | | | |
|---------|----|-----|-----|-----|-----|------|-----|-----|-----|-----|-----|------|-----|-----|-----|-----|-----|------|
| Proportion of variables retained | | 5% | 10% | 15% | 20% | 50% | 100% | 5% | 10% | 15% | 20% | 50% | 100% | 5% | 10% | 15% | 20% | 30% | 100% |
| iTransformer | NFS | 0.201 | 0.185 | 0.180 | 0.177 | 0.167 | 0.148 | 0.260 | 0.248 | 0.227 | 0.222 | 0.214 | 0.206 | 0.428 | 0.408 | 0.402 | 0.399 | 0.397 | 0.395 |
| | Influence selection | 0.187 | 0.174 | 0.170 | 0.165 | 0.150 | 0.148 | 0.229 | 0.224 | 0.220 | 0.219 | 0.210 | 0.206 | 0.419 | 0.405 | 0.398 | 0.397 | 0.395 | 0.395 |

From the results shown in the table, it is evident that our method is more effective. According to the method described in the paper (Gu et al., 2021), this approach introduces additional network parameters to evaluate the importance of different channels. Furthermore, the number of additional parameters required by this method scales with the number of channels, significantly increasing its computational time. Specifically, while the original iTransformer takes only 17 seconds to train one epoch on the ECL dataset, this method increases the time to 32 seconds per epoch.

