# OpenReview forum: "Channel-wise Influence: Estimating Data Influence for Multivariate Time Series"
_ICLR.cc/2025/Conference — ICLR 2025 Conference Withdrawn Submission_

### Official Review · Reviewer_4y6T · 2024-11-01

**Soundness:** 3
**Presentation:** 3
**Contribution:** 3
**Rating:** 6
**Confidence:** 4

**Summary:**

The paper introduces a new channel-wise influence function designed to address limitations in multivariate time series (MTS) analysis. Unlike previous model-centric methods, this approach provides a data-centric view, allowing the estimation of the influence of each channel within the MTS. The authors propose an influence function leveraging a first-order gradient approximation to improve MTS anomaly detection and forecasting tasks. Experimental results on various datasets indicate that the method outperforms traditional influence functions, especially in anomaly detection and channel pruning for forecasting.

**Strengths:**

Originality: Proposes a unique channel-wise influence function, filling an important gap in MTS analysis by focusing on data-centric rather than model-centric approaches.

Quality:Demonstrates robust experiments across datasets, highlighting improvements in MTS anomaly detection and forecasting.

Clarity: Clear, structured explanations and well-supported claims for the impact of the influence function on downstream MTS tasks.

Significance: The model holds practical relevance across domains where MTS is essential, particularly in applications needing channel-wise insights.

**Weaknesses:**

1. Figure 1 is not cited within the paper.
2. Algorithm 2 could be polished.

**Questions:**

1. After channel pruning, does the final prediction use the pruned channels or the full set of channels? Clarifying this will help determine whether pruning directly improves prediction or simply reduces computational costs, and why are the pruned channels still accurately predicted? How does the model achieve this?

2. Given that PatchTST is channel-independent, how can we justify that removing channels enhances overall performance? For example, certain channels may have lower MSE due to higher predictability, while others may be more erratic, resulting in higher MSE. If more erratic channels are removed, this might artificially lower the overall MSE, which could be misleading.

3. Rather than pruning channels, would it be more insightful to analyze the interactions between channels? For example, investigating how one channel affects another could provide a deeper understanding of channel dependencies in MTS.

4. Real-world MTS often exhibit dynamic relationships between channels, which may vary over time. For instance, two channels might show positive correlation at one point and no correlation at another. Could pruning lead to a loss of such dynamic, context-dependent information?

5. From the results in Table 5, it appears that the pruned models only slightly outperform or match the full-channel models. This raises questions about the necessity and practical benefits of pruning. How does channel pruning substantively benefit the analysis or forecasting tasks, given these marginal differences?

6. Based on the above Q1 and Q2, the biggest confusion is：we think all comparison experiments should maintain consistent output channels (i.e., the same number of channels) to ensure fairness and accuracy when evaluating the model's performance?

---

> ### Author Response · Authors · 2024-11-15
> **Explanation of the contributions of the paper:**
>
> We appreciate your time to provide valuable comments and suggestions to improve our paper substantially. Before we begin addressing your questions, we would like to first clarify the primary contributions of our paper. Influence functions have demonstrated significant performance and value across various fields[4]-[10], with notable applications in outlier detection[7][8][9][10] and data pruning[4][5]. However, there is no preceding research on the influence functions of multivariate time series to shed light on the effects of modifying the channel of multivariate time series. To fill this gap, we propose a channel-wise influence function, which is the first data-centric method that can estimate the influence of different channels in multivariate time series. We conducted extensive experiments and found that the original influence function performed poorly in anomaly detection and could not facilitate channel pruning, underscoring the superiority and necessity of our approach. Additionally, we can further analyze the information learned by the model through the channel-wise influence matrix. For example, as mentioned in Section 5.2.1 of our paper, comparing the number of core channel subsets across different models reveals each model's capacity for capturing channel dependencies. A larger core subset indicates that the model has captured more effective information, which also highlights the interpretability of our approach.
>
> **Regarding the selection of tasks to verify our method:** Anomaly detection has long been a critical issue in multivariate time series analysis, with relevant studies including[1][2]. Data pruning is equally important as it raises the question of whether we can train a high-performing model with less data than predicted by scaling laws[3], with related work found in studies such as [3][4][5]. However, data pruning has not yet been extensively studied within the context of time series. By analyzing the characteristics of multivariate time series, we identified redundancy between channels and developed a channel pruning method based on our proposed channel-wise influence function, which outperforms traditional data pruning approaches. In addition, this method also supports the interpretability analysis of the model's ability to model multivariate time series.
>
> [1]Position paper: Quo vadis, unsupervised time series anomaly detection? ICML 2024
>
> [2]Anomaly transformer: Time series anomaly detection with association discrepancy. ICLR 2022
>
> [3]Beyond neural scaling laws: beating power law scaling via data pruning Neurips 2022
>
> [4]Data Pruning via Moving-one-Sample-out Neurips 2023
>
> [5]Dataset Pruning: Reducing Training Data by Examining Generalization Influence ICLR 2023
>
> [6]Self-influence guided data reweighting for language model pre-training. ACL 2023
>
> [7]Detecting adversarial samples using influence functions and nearest neighbors. CVPR 2020
>
> [8]Estimating training data influence by tracing gradient descent. Neurips 2020
>
> [9]Understanding Black-box Predictions via Influence Functions. ICML 2017
>
> [10]Resolving Training Biases via Influence-based Data Relabeling. ICLR 2022

---

> > ### Author Response · Authors · 2024-11-15
> > **Response to weaknesses**
> >
> > **Q1:** Figure 1 is not cited within the paper.
> >
> > **A1：** We have revised the paper according to your suggestions and re-uploaded it. The modified sections are highlighted in blue for your reference.
> >
> > **Q2:** Algorithm 2 could be polished.
> >
> > **A2：** We have revised the paper according to your suggestions and re-uploaded it. The modified sections are highlighted in blue for your reference. In addition, we would appreciate any further suggestions you may have for refining Algorithm 2. This would greatly assist us in better explaining our work to you.

---

> > > ### Author Response · Authors · 2024-11-15
> > > **Response to Questions**
> > >
> > > **Q1:** After channel pruning, does the final prediction use the pruned channels or the full set of channels? Clarifying this will help determine whether pruning directly improves prediction or simply reduces computational costs, and why are the pruned channels still accurately predicted? How does the model achieve this?
> > >
> > > **A1：** To ensure a fair comparison, the model’s output during testing includes predictions for all channels. The main goals of channel pruning are twofold: first, by removing redundant channel information in the training set, we can accelerate model training; second, channel pruning allows us to identify an effective training channel subset for a specific model, supporting interpretability analysis. A larger subset (The proportion marked in red in Table 5 represents the size of the core subset.) indicates a stronger capability of the model to capture channel dependencies in time series data. The fact that pruned channels can still be accurately predicted is due to the presence of channels with similar patterns to those removed during training, revealing redundancy between channels. This redundancy enables the model to learn one channel’s trend and generalize it effectively to accurately predict other channels.
> > >
> > >
> > >
> > > **Q2:** Given that PatchTST is channel-independent, how can we justify that removing channels enhances overall performance? For example, certain channels may have lower MSE due to higher predictability, while others may be more erratic, resulting in higher MSE. If more erratic channels are removed, this might artificially lower the overall MSE, which could be misleading.
> > >
> > > **A2：** Referring to the explanation in Q1, we did not test only a subset of the channels, but instead evaluated the prediction results for all channels. Therefore, you don't need to worry about this case.
> > >
> > >
> > > **Q3:** Rather than pruning channels, would it be more insightful to analyze the interactions between channels? For example, investigating how one channel affects another could provide a deeper understanding of channel dependencies in MTS.
> > >
> > > **A3：** Thank you for your suggestion. We have addressed this issue in Remark 3.2 of the paper. We believe that the channel-wise influence matrix indeed reflects how different models understand channel correlations. This is why iTransformer has a larger core subset—it can more accurately distinguish and identify the relationships between different channels. This also explains why we conducted this experiment, as the core subset reflects whether the model is capable of understanding which channels are related and different from each other.
> > >
> > > **Q4:** Real-world MTS often exhibit dynamic relationships between channels, which may vary over time. For instance, two channels might show positive correlation at one point and no correlation at another. Could pruning lead to a loss of such dynamic, context-dependent information?
> > >
> > > **A4：** Thank you for your suggestion. Since we are focusing on the channels that are more important to the model's performance throughout the entire training process, rather than at a specific moment, it does not reflect the dynamic channel correlations. Addressing this issue could be a new research direction we may consider moving forward.
> > >
> > > **Q5:** From the results in Table 5, it appears that the pruned models only slightly outperform or match the full-channel models. This raises questions about the necessity and practical benefits of pruning. How does channel pruning substantively benefit the analysis or forecasting tasks, given these marginal differences?
> > >
> > > **A5：** Referring to our previous response to Q1, the purpose of pruning is not to improve model performance, but rather to accelerate training and facilitate interpretability analysis of the model's capabilities. If I have misunderstood your question, please feel free to clarify, and I would be happy to discuss it further.
> > >
> > > **Q6:** Based on the above Q1 and Q2, the biggest confusion is：we think all comparison experiments should maintain consistent output channels (i.e., the same number of channels) to ensure fairness and accuracy when evaluating the model's performance?
> > >
> > > **A6：** Referring to our previous response to Q1, we predict the results for all channels during testing, so the comparison is valid. If I have misunderstood your question, please feel free to clarify, and I would be happy to discuss it further.

---

> > > > ### Author Response · Authors · 2024-11-24
> > > > **Gentle reminder**
> > > >
> > > > Dear Reviewer  4y6T
> > > >
> > > > We sincerely appreciate your time and effort in reviewing our manuscript and offering valuable suggestions. As the discussion time is coming to an end, we would like to confirm whether our responses have effectively addressed your concerns. We provided detailed responses to your concerns a few days ago, and we hope they have adequately addressed your issues. If you require further clarification or have any additional concerns, please do not hesitate to contact us. We are more than willing to continue our communication with you.
> > > >
> > > > Best regards,

---

> > > > ### Comment · Reviewer_4y6T · 2024-11-27
> > > > **Thank you for your response**
> > > >
> > > > Thank you for your response. Our questions are well-answered. I have raised the score to 6.

---

> > > > > ### Author Response · Authors · 2024-11-27
> > > > > **Thank you for your response**
> > > > >
> > > > > We are glad that we have addressed your questions. Thank you for your valuable suggestions and time. If you have any new questions, we would be glad to have a further discussion with you.

---

### Official Review · Reviewer_3pTH · 2024-11-02

**Soundness:** 2
**Presentation:** 2
**Contribution:** 2
**Rating:** 5
**Confidence:** 4

**Summary:**

This paper introduces the Channel-wise Influence Function, a method designed to analyze the influence of individual channels in MTS data on the performance of ML models. The authors argue that existing influence function methods, commonly applied in CV and NLP, are inadequate for MTS analysis due to temporal dependencies and diverse information contained within different channels of MTS data. The proposed method leverages a first-order gradient approximation, drawing inspiration from the TracIn method, to quantify how training with a specific channel in the MTS data affects the model's ability to predict another channel. This method is presented to provide insights into the relationship between data and model behavior.

**Strengths:**

(1) The proposed method addresses a limitation in existing influence function methods by studying the unique contributions of individual channels within MTS data. While traditional methods assess the impact of entire data samples on model performance.
(2) By ranking channels based on their self-influence scores, the proposed method enables the selection of a reduced subset of channels without significant compromise to the model's predictive accuracy. This is particularly advantageous in scenarios where training with the entire set of channels is computationally expensive or infeasible.

**Weaknesses:**

(1) The paper primarily focuses on anomaly detection and forecasting, leaving the application to other relevant MTS tasks, such as classification, clustering, or imputation, unexplored. This limited scope restricts the paper's ability to fully demonstrate the potential of the proposed method in diverse MTS applications.
(2) The method relies on gradient computations, which can become computationally demanding for complex models, particularly when applied to large-scale MTS datasets. To address this, the paper proposes using gradients from a subset of model parameters to improve efficiency. However, a more detailed analysis of the trade-off between computational cost and performance when employing a reduced set of gradients is warranted.
(3) The paper employs an equidistant sampling strategy to select channels based on their ranked self-influence scores. This approach may introduce biases, particularly when the distribution of influence scores is uneven.  For instance, if a large number of channels have similar influence scores, the equidistant sampling might lead to the exclusion of potentially informative channels simply because they are clustered together in the ranking.

**Questions:**

(1) Several works have successfully applied influence functions to analyze MTS data. For instance, TimeInf (Li et al., 2024) utilizes influence functions to quantify the impact of individual data points on the model's predictions. It would be beneficial for the authors to explicitly discuss how the proposed method compares to, or builds upon, these existing approaches.

1. TimeInf: Time Series Data Contribution via Influence Functions. https://arxiv.org/abs/2407.15247
2. Interdependency Matters: Graph Alignment for Multivariate Time Series Anomaly Detection. https://arxiv.org/abs/2410.08877
3. sTransformer: A Modular Approach for Extracting Inter-Sequential and Temporal Information for Time-Series Forecasting. https://arxiv.org/abs/2408.09723

(2) The paper primarily focuses on comparing with the original influence function and naive channel selection methods. A more comprehensive evaluation would involve comparing with other channel selection techniques, such as those based on feature importance scores or attention mechanisms, to provide a more robust assessment of its effectiveness.

(3) Computational complexity analysis of the proposed method as the number of channels increases  is crucial. Hence, an empirical evaluation using larger and more diverse datasets, such as the publicly available ETT, Electricity, and Traffic datasets (link: https://drive.google.com/file/d/1l51QsKvQPcqILT3DwfjCgx8Dsg2rpjot/view), would be beneficial.

(4) While the paper emphasizes the potential of the proposed method for post-hoc model analysis, particularly in the context of channel pruning, it is worth exploring further applications. For instance, could metrics derived from the channel influence matrix, such as entropy or diversity of influential channels, be leveraged to compare and evaluate different MTS models? Such metrics might provide insights into a model’s ability to capture complex channel relationships and its overall performance.

(5) For the selection of representative channels, I'm wondering if methods like top-k selection or adaptive sampling based on the influence score distribution be more appropriate?

(6) The 1-Layer MLP model in Table 3 appears to refer to a single-layer Multilayer Perceptron as the entire model architecture. It is intriguing that such a simple model can achieve comparable, and in some cases, superior performance to a more complex Transformer-based model?

---

> ### Author Response · Authors · 2024-11-17
> **Explanation of the contributions of the paper:**
>
> We appreciate your time to provide valuable comments and suggestions to improve our paper substantially. Before we begin addressing your questions, we would like to first clarify the primary contributions of our paper. Influence functions have demonstrated significant performance and value across various fields[4]-[10], with notable applications in outlier detection[7][8][9][10] and data pruning[4][5]. However, there is no preceding research on the influence functions of multivariate time series to shed light on the effects of modifying the channel of multivariate time series. To fill this gap, we propose a channel-wise influence function, which is the first data-centric method that can estimate the influence of different channels in multivariate time series. We conducted extensive experiments and found that the original influence function performed poorly in anomaly detection and could not facilitate channel pruning, underscoring the superiority and necessity of our approach. Additionally, we can further analyze the information learned by the model through the channel-wise influence matrix. For example, as mentioned in Section 5.2.1 of our paper, comparing the number of core channel subsets across different models reveals each model's capacity for capturing channel dependencies. A larger core subset indicates that the model has captured more effective information, which also highlights the interpretability of our approach.
>
> **Regarding the selection of tasks to verify our method:** Anomaly detection has long been a critical issue in multivariate time series analysis, with relevant studies including[1][2]. Data pruning is equally important as it raises the question of whether we can train a high-performing model with less data than predicted by scaling laws[3], with related work found in studies such as [3][4][5]. However, data pruning has not yet been extensively studied within the context of time series. By analyzing the characteristics of multivariate time series, we identified redundancy between channels and developed a channel pruning method based on our proposed channel-wise influence function, which outperforms traditional data pruning approaches. In addition, this method also supports the interpretability analysis of the model's ability to model multivariate time series.
>
> [1]Position paper: Quo vadis, unsupervised time series anomaly detection? ICML 2024
>
> [2]Anomaly transformer: Time series anomaly detection with association discrepancy. ICLR 2022
>
> [3]Beyond neural scaling laws: beating power law scaling via data pruning Neurips 2022
>
> [4]Data Pruning via Moving-one-Sample-out Neurips 2023
>
> [5]Dataset Pruning: Reducing Training Data by Examining Generalization Influence ICLR 2023
>
> [6]Self-influence guided data reweighting for language model pre-training. ACL 2023
>
> [7]Detecting adversarial samples using influence functions and nearest neighbors. CVPR 2020
>
> [8]Estimating training data influence by tracing gradient descent. Neurips 2020
>
> [9]Understanding Black-box Predictions via Influence Functions. ICML 2017
>
> [10]Resolving Training Biases via Influence-based Data Relabeling. ICLR 2022

---

> ### Author Response · Authors · 2024-11-17
> **Response to weaknesses**
>
> **Q1:** The paper primarily focuses on anomaly detection and forecasting, leaving the application to other relevant MTS tasks, such as classification, clustering, or imputation, unexplored. This limited scope restricts the paper's ability to fully demonstrate the potential of the proposed method in diverse MTS applications.
>
> **A1：** Considering that the contribution of our paper lies in introducing a data-centric approach to evaluate channel-wise influence for multivariate time series, it is not feasible to validate our method across all time series methods. Extending our approach to other application tasks is a direction for future research, which we have acknowledged in the limitations section of the paper. Respectly, we do not view this as a weakness, as we have thoroughly demonstrated the effectiveness of our data-centric method in two representative time series tasks.
>
> **Q2:** The method relies on gradient computations, which can become computationally demanding for complex models, particularly when applied to large-scale MTS datasets. To address this, the paper proposes using gradients from a subset of model parameters to improve efficiency. However, a more detailed analysis of the trade-off between computational cost and performance when employing a reduced set of gradients is warranted.
>
> **A2：** In Section 5.1.3 of our paper, *Parameter Analysis*, we thoroughly discussed the impact of the number of gradient parameters used in the computation across three different datasets and found no significant differences. Additionally, other studies on influence functions have also addressed this trade-off, as referenced in [1], [2], and [3]. These works have highlighted that using only a subset of the model’s parameters can still achieve excellent results. Therefore, we believe that accelerating the computation of influence by calculating gradients for only a portion of the parameters is a reasonable and effective approach.
>
> **Q3:** The paper employs an equidistant sampling strategy to select channels based on their ranked self-influence scores. This approach may introduce biases, particularly when the distribution of influence scores is uneven. For instance, if a large number of channels have similar influence scores, the equidistant sampling might lead to the exclusion of potentially informative channels simply because they are clustered together in the ranking.
>
> **A3：** Thank you very much for your suggestion. We have also considered a similar issue, and a detailed response can be found in *Response to Questions*, Q5.
>
> [1]Dataset Pruning: Reducing Training Data by Examining Generalization Influence ICLR 2023
>
> [2]Self-influence guided data reweighting for language model pre-training. ACL 2023
>
> [3]Estimating training data influence by tracing gradient descent. Neurips 2020

---

> > ### Author Response · Authors · 2024-11-17
> > **Response to Questions:**
> >
> > **Q1:** Several works have successfully applied influence functions to analyze MTS data. For instance, TimeInf (Li et al., 2024) utilizes influence functions to quantify the impact of individual data points on the model's predictions. It would be beneficial for the authors to explicitly discuss how the proposed method compares to, or builds upon, these existing approaches.
> >
> > TimeInf: Time Series Data Contribution via Influence Functions. https://arxiv.org/abs/2407.15247
> > Interdependency Matters: Graph Alignment for Multivariate Time Series Anomaly Detection. https://arxiv.org/abs/2410.08877
> > sTransformer: A Modular Approach for Extracting Inter-Sequential and Temporal Information for Time-Series Forecasting. https://arxiv.org/abs/2408.09723
> >
> > **A1：** Thank you for providing the references. After careful review, we believe that only the first article specifically addresses the use of influence functions in time series. To be honest, when we completed our work, these articles did not exist on the Internet.Since all three articles were published after July, we consider them to be concurrent work. Therefore, we believe it is reasonable not to include comparisons with these works in our study.
> >
> > In addition, the specific distinctions between our work and the referenced papers are as follows:
> >
> > **Difference from TimeInf**: TimeInf is an influence function inspired by autoregressive models, primarily focused on autoregressive settings and motivated by the goal of more accurately calculating influence functions while considering temporal dependencies. In contrast, our approach is the first to focus on channel-wise influence evaluation. We emphasize evaluating the influence among channels rather than across time steps. Additionally, our channel-wise influence function effectively produces a channel-wise influence matrix, which captures the inter-channel relationships during training. This matrix can be utilized to derive the core channel subset for training, further aiding in the interpretability of the model's capabilities.
> >
> > **Differences from the other two methods**: The latter two approaches are task-specific, model-centric methods designed to achieve better performance in specific tasks. This is fundamentally different from our data-centric approach and contributions. For details on our contributions, please refer to *Explanation of the Contributions of the Paper*.
> >
> > **Q2:** The paper primarily focuses on comparing with the original influence function and naive channel selection methods. A more comprehensive evaluation would involve comparing with other channel selection techniques, such as those based on feature importance scores or attention mechanisms, to provide a more robust assessment of its effectiveness.
> >
> > **A2：** Since we are the first to propose using channel pruning instead of data pruning for multivariate time series tasks, there are no existing baselines of this type available for direct comparison. Therefore, we consider this an area for future research. If possible, could you kindly provide specific references or articles that we can compare against?
> >
> > **Q3:** Computational complexity analysis of the proposed method as the number of channels increases is crucial. Hence, an empirical evaluation using larger and more diverse datasets, such as the publicly available ETT, Electricity, and Traffic datasets (link: https://drive.google.com/file/d/1l51QsKvQPcqILT3DwfjCgx8Dsg2rpjot/view), would be beneficial.
> >
> > **A3：** In our experiments, we utilized the ETTm1, Solar-Energy, Electricity, and Traffic datasets to analyze the computational complexity of our method. By measuring the time required for calculating single-instance influence, we demonstrated how the computational time scales with the number of channels.
> >
> > |              | ETTm1        | Solar-Energy| Electricity | traffic |
> > |--------------|--------------|-------------|---------|---------|
> > | iTransformer+ours |     0.0025s         |       0.023s      |   0.071s      |    0.18s     |
> >
> > From the table, it can be observed that the computational complexity approximately increases linearly with the number of channels.

---

> > > ### Author Response · Authors · 2024-11-17
> > > **Response to Questions:**
> > >
> > > **Q4:** While the paper emphasizes the potential of the proposed method for post-hoc model analysis, particularly in the context of channel pruning, it is worth exploring further applications. For instance, could metrics derived from the channel influence matrix, such as entropy or diversity of influential channels, be leveraged to compare and evaluate different MTS models? Such metrics might provide insights into a model’s ability to capture complex channel relationships and its overall performance.
> > >
> > > **A4** Thank you for your suggestion. Regarding your idea of directly deriving evaluation metrics from the influence matrix, we think it's an excellent approach. However, after some initial consideration, we believe that metrics like entropy and diversity of the matrix can only measure the channel utilization ratio of the model, but they do not accurately reflect the model's effectiveness in utilizing those channels. To illustrate this with a simple example: if a model's channel-wise influence matrix has high entropy, it means the model has attended to all channels. However, if the model hasn't learned much from these channels, meaning the influence across all channels is low, the actual test loss might still be high. Therefore, we believe that our approach, which involves using channel pruning to identify the core training subset and evaluating the model's channel modeling capability based on the size of the core training subset, provides a more accurate assessment of the model's ability.
> > > In Section 5.2.1 of the paper, under Result Analysis and Outlook, we conducted an interpretability analysis of the model's capability. Specifically, the size of the core subset reflects the model's utilization of information across different channels. If a model has a larger core subset (indicated by the red-highlighted proportions in Table 5), it suggests that the model can effectively distinguish and capture the relationships between different channels. This is because it leverages the information from various channels to improve its predictions. The more information utilized, the stronger the model's ability to capture channel dependencies. Therefore, the size of the core subset provides an explanation of the model's capacity to model channel dependencies. In other words, the size of the core subset can also serve as a metric to assess this capability.
> > >
> > > **Q5:** For the selection of representative channels, I'm wondering if methods like top-k selection or adaptive sampling based on the influence score distribution be more appropriate?
> > >
> > > **A5：** Thank you for your valuable suggestion. In *Appendix C.1*, we compared the top-k selection method, which involves selecting the most influential samples. The results indicate that this approach does not perform as well as equidistant sampling. We attribute this to the fact that channels with high influence may sometimes be outliers in terms of data distribution or represent channels that are particularly difficult to predict (e.g., channels with extreme variations). Overemphasizing such high-influence channels may not necessarily lead to better pruning results.
> > >
> > > The rationale behind our use of equidistant sampling is to maximize channel diversity retention. While we have not conducted an in-depth study on adaptive sampling methods, we consider this an exciting avenue for future research. In addition, respectly, we don't think this problem can be a weakness, because there is always a new method to improve the previous method. If you would like to share further insights or suggestions, we would be eager to learn from them.
> > >
> > > **Q6:** The 1-Layer MLP model in Table 3 appears to refer to a single-layer Multilayer Perceptron as the entire model architecture. It is intriguing that such a simple model can achieve comparable, and in some cases, superior performance to a more complex Transformer-based model?
> > >
> > > **A6：** The results in Table 3 are directly copied from the ICML 2024 paper[1], which represents the core argument of that study—namely, that modern deep learning models are, in some cases, merely presenting a façade of progress and do not necessarily outperform simpler machine learning models.
> > >
> > > [1]Position paper: Quo vadis, unsupervised time series anomaly detection? ICML 2024

---

> > > > ### Author Response · Authors · 2024-11-24
> > > > **Gentle reminder**
> > > >
> > > > Dear Reviewer 3pTH
> > > >
> > > > We sincerely appreciate your time and effort in reviewing our manuscript and offering valuable suggestions. As the discussion time is coming to an end, we would like to confirm whether our responses have effectively addressed your concerns. We provided detailed responses to your concerns a few days ago, and we hope they have adequately addressed your issues. If you require further clarification or have any additional concerns, please do not hesitate to contact us. We are more than willing to continue our communication with you.
> > > >
> > > > Best regards,

---

> ### Comment · Reviewer_3pTH · 2024-11-25
>
> I appreciate the author's efforts to address my previous concerns. However, a few issues remain:
>
> 1. The revised version does not appear to include the additional experiments. For example those utilizing the ETTh1, ETTm1 datasets, and the DLinear model, etc.
>
> 2. Regarding baselines comparison, I think there might be some relevant methods to compare to such those using either channel independence (CI) or channel dependence (CD) for time series analysis. Following are some examples that might be relevant:
>
> [1] Rethinking Channel Dependence for Multivariate Time Series Forecasting: Learning from Leading Indicators. ICLR 2024
>
> [2] Feature Selection for Multivariate Time Series via Network Pruning. https://arxiv.org/pdf/2102.06024
>
> 3. My previous request regarding computational complexity remains open. I require an analysis of the proposed method's computational complexity, particularly its scalability with high-dimensional data and the trade-off between this complexity and the achieved performance gains and not the complexity for a single channel.
>
> 4. Regarding your answer to Q6: The observation that a simple MLP model with only one layer outperforms a Transformer model raises concerns about the chosen datasets. It is possible that the datasets used might lack the complexity typically observed in real-world multivariate time series (MTS) data, where intricate temporal dynamics are prevalent.
>
> However, I believe a rating of 5 is more appropriate. I would like to adjust my previous score of 3 accordingly.

---

> > ### Author Response · Authors · 2024-11-25
> > **Reponse to new issuses**
> >
> > Thank you for your detail responses. In response to your new questions, here are our new answers.
> > If we have misunderstood your question in any way, we kindly ask for further discussion.
> >
> > **Q1:** The revised version does not appear to include the additional experiments. For example those utilizing the ETTh1, ETTm1 datasets, and the DLinear model, etc.
> >
> > **A1:** Thank you for your suggestion. We have added the new experiments in the appendix C.3 and C.4, and uploaded the revised version of the paper for your review.
> >
> > **Q2:** Regarding baselines comparison, I think there might be some relevant methods to compare to such those using either channel independence (CI) or channel dependence (CD) for time series analysis. Following are some examples that might be relevant:
> >
> > [1] Rethinking Channel Dependence for Multivariate Time Series Forecasting: Learning from Leading Indicators. ICLR 2024
> >
> > [2] Feature Selection for Multivariate Time Series via Network Pruning. https://arxiv.org/pdf/2102.06024
> >
> > **A2:**  Thank you for your feedback. As far as we know, DLinear is also a very representative channel-independence method because it assumes that all channels share the same linear parameters, as mentioned in [1]. That explains why we choose this model. Regarding the first paper you provided, it is a plug-and-play method that can be applied to various models, so we are currently considering how to make a fair comparison with our method. If you have any suggestions, we would be happy to hear them. Lastly, concerning the second paper, since we couldn't find the authors' open-source code, we are attempting to reproduce their results.
> >
> > [1] A TIME SERIES IS WORTH 64 WORDS: LONG-TERM FORECASTING WITH TRANSFORMERS ICLR 2023
> >
> > **Q3：** My previous request regarding computational complexity remains open. I require an analysis of the proposed method's computational complexity, particularly its scalability with high-dimensional data and the trade-off between this complexity and the achieved performance gains and not the complexity for a single channel.
> >
> > **A3:**  We think that there may be some misunderstanding regarding our experiments. The experiments conducted in Appendix C.4 measure the influence for the entire multivariate time series, meaning we calculated the influence across all channels rather than only a single channel. Specifically, we computed the influence for a single MTS data sample, which includes a large number of channels.Furthermore, we are not entirely sure we understand your point regarding the trade-off between high-dimensional data and the achieved performance gains, as we always compute the influence of all channels within a single MTS data sample. Thus, such a trade-off should not exist in our case. If we have misunderstood your question, please feel free to clarify.
> >
> > **Q4：** Regarding your answer to Q6: The observation that a simple MLP model with only one layer outperforms a Transformer model raises concerns about the chosen datasets. It is possible that the datasets used might lack the complexity typically observed in real-world multivariate time series (MTS) data, where intricate temporal dynamics are prevalent.
> >
> > **A4:** Thank you for your suggestion. This aligns with one of the concerns mentioned in the paper[2], namely that existing datasets for time series anomaly detection might be somewhat simplistic. This is also one of the reasons to explain why we used MTS forecasting datasets in the channel pruning task to further demonstrate the effectiveness of our method.
> >
> > [2]Position paper: Quo vadis, unsupervised time series anomaly detection? ICML 2024

---

> ### Author Response · Authors · 2024-11-25
> **Additional Repsonse to Q2:**
>
> To better highlight the effectiveness of our method, we compared it with the approach proposed in the mentioned paper[1], referred to as NFS. The specific results are as follows:
>
> |              Dataset             |                     |  ECL  |       |       |       |       |       | Solar |       |       |       |       |       | Traffic |       |       |       |       |       |
> |:--------------------------------:|:-------------------:|:-----:|:-----:|:-----:|:-----:|:-----:|:-----:|:-----:|:-----:|:-----:|:-----:|:-----:|:-----:|:-------:|:-----:|:-----:|:-----:|:-----:|:-----:|
> | Proportion of variables retained |                     |   5%  |  10%  |  15%  |  20%  |  50%  |  100% |   5%  |  10%  |  15%  |  20%  |  50%  |  100% |    5%   |  10%  |  15%  |  20%  |  30%  |  100% |
> |            iTransformer          |         NFS         | 0.201 | 0.185 | 0.180 | 0.177 | 0.167 | 0.148 | 0.260 | 0.248 | 0.227 | 0.222 | 0.214 | 0.206 |  0.428  | 0.408 | 0.402 | 0.399 | 0.397 | 0.395 |
> |                                  | Influence selection | 0.187 | 0.174 | 0.170 | 0.165 | 0.150 | 0.148 | 0.229 | 0.224 | 0.220 | 0.219 | 0.210 | 0.206 |  0.419  | 0.405 | 0.398 | 0.397 | 0.395 | 0.395 |
>
> From the results shown in the table, it is evident that our method is more effective. According to the method described in the paper[1], this approach introduces additional network parameters to evaluate the importance of different channels. Furthermore, the number of additional parameters required by this method scales with the number of channels, significantly increasing its computational time. Specifically, while the original iTransformer takes only 17 seconds to train one epoch on the ECL dataset, this method increases the time to 32 seconds per epoch.

---

> > ### Author Response · Authors · 2024-12-02
> > **Gentle Reminder**
> >
> > Dear Reviewer 3pTH,
> >
> > We sincerely appreciate your time and effort in reviewing our manuscript and offering valuable suggestions. As the discussion phase is drawing to an end, we sincerely hope to know whether we have addressed all of your concerns. If you have any further questions, we would be glad to engage in further discussion.
> >
> > Best Regards

---

### Official Review · Reviewer_eNTa · 2024-11-04

**Soundness:** 3
**Presentation:** 2
**Contribution:** 3
**Rating:** 6
**Confidence:** 3

**Summary:**

This paper studies the problem of influence function for multivariate time series (MTS), which is the first study of MTS in deep learning. To effectively estimate the influence of MTS, this paper proposes a first-order gradient approximation. Then, the authors propose two channel-wise influence function-based algorithms for MTS anomaly detection and forecasting, respectively.

**Strengths:**

- It is the first work of influence function for MTS in deep learning.

- Two channel-wise influence function-based algorithms is proposed in this paper to be applied in MTS anomaly detection and forecasting tasks.

**Weaknesses:**

- The technical contribution of this paper is not very high. Only the influence function is proposed in MTS, which has been well-studied in other domains.

- The experimental results are not very impressive. In Table 2, we can observe that by using of proposed influence the improvement is not very significant. And also the datasets used in this paper are not enough.

- The time complexity of the proposed method is not analyzed in this paper.

- The code of this paper is not provided.

**Questions:**

See above weaknesses.

---

> ### Author Response · Authors · 2024-11-17
> **Response to weaknesses:**
>
> We appreciate your time to provide valuable comments and suggestions to improve our paper substantially.
>
> **Q1:** The technical contribution of this paper is not very high. Only the influence function is proposed in MTS, which has been well-studied in other domains.
> **A1:** Before we begin addressing your questions, we would like to first clarify the primary contributions of our paper. Influence functions have demonstrated significant performance and value across various fields[4]-[10], with notable applications in outlier detection[7][8][9][10] and data pruning[4][5]. However, there is no preceding research on the influence functions of multivariate time series to shed light on the effects of modifying the channel of multivariate time series. To fill this gap, we propose a channel-wise influence function, which is the first data-centric method that can estimate the influence of different channels in multivariate time series. We conducted extensive experiments and found that the original influence function performed poorly in anomaly detection and could not facilitate channel pruning, underscoring the superiority and necessity of our approach. Additionally, we can further analyze the information learned by the model through the channel-wise influence matrix. For example, as mentioned in Section 5.2.1 of our paper, comparing the number of core channel subsets across different models reveals each model's capacity for capturing channel dependencies. A larger core subset indicates that the model has captured more effective information, which also highlights the interpretability of our approach.
>
> **Regarding the selection of tasks to verify our method:** Anomaly detection has long been a critical issue in multivariate time series analysis, with relevant studies including[1][2]. Data pruning is equally important as it raises the question of whether we can train a high-performing model with less data than predicted by scaling laws[3], with related work found in studies such as [3][4][5]. However, data pruning has not yet been extensively studied within the context of time series. By analyzing the characteristics of multivariate time series, we identified redundancy between channels and developed a channel pruning method based on our proposed channel-wise influence function, which outperforms traditional data pruning approaches. In addition, this method also supports the interpretability analysis of the model's ability to model multivariate time series.
>
> [1]Position paper: Quo vadis, unsupervised time series anomaly detection? ICML 2024
>
> [2]Anomaly transformer: Time series anomaly detection with association discrepancy. ICLR 2022
>
> [3]Beyond neural scaling laws: beating power law scaling via data pruning Neurips 2022
>
> [4]Data Pruning via Moving-one-Sample-out Neurips 2023
>
> [5]Dataset Pruning: Reducing Training Data by Examining Generalization Influence ICLR 2023
>
> [6]Self-influence guided data reweighting for language model pre-training. ACL 2023
>
> [7]Detecting adversarial samples using influence functions and nearest neighbors. CVPR 2020
>
> [8]Estimating training data influence by tracing gradient descent. Neurips 2020
>
> [9]Understanding Black-box Predictions via Influence Functions. ICML 2017
>
> [10]Resolving Training Biases via Influence-based Data Relabeling. ICLR 2022

---

> ### Author Response · Authors · 2024-11-17
> **Response to weaknesses:**
>
> **Q2:** The experimental results are not very impressive. In Table 2, we can observe that by using of proposed influence the improvement is not very significant. And also the datasets used in this paper are not enough.
>
> **A2:** Our method achieved a 7% improvement on the SMD dataset and a 15% improvement on the SMAP dataset, along with varying degrees of improvement on other anomaly detection datasets, demonstrating its effectiveness. For the time-series anomaly detection task, we utilized five real-world datasets, while for the time-series forecasting task, we employed three real-world datasets. Additionally, we added experiments on the ETTh1 and ETTm1 datasets, as shown in the table below. Overall, we believe our use of a relatively large number of datasets in time-series research highlights the generalizability and effectiveness of our approach.
>
> Since the original number of channels in ETTh1 and ETTm1 is only 7, the horizontal axis in the table directly represents the number of retained channels.
>
> |              Dataset             |                      | ETTh1 |       |       | ETTm1 |       |       |
> |:--------------------------------:|:--------------------:|:-----:|:-----:|:-----:|:-----:|:-----:|:-----:|
> | number of channels retained |                      |   7   |   3   |   2   |   7   |   3   |   2   |
> |           iTransformer           | Continuous selection | 0.396 | 0.502 | 0.573 | 0.332 | 0.756 | 0.826 |
> |                                  |   Random selection   | 0.396 | 0.428 | 0.434 | 0.332 | 0.362 | 0.372 |
> |                                  |  Influence selection | 0.396 | 0.403 | 0.420 | 0.332 | 0.333 | 0.355 |
> |             PatchTST             | Continuous selection | 0.400 | 0.460 | 0.491 | 0.330 | 0.539 | 0.687 |
> |                                  |   Random selection   | 0.400 | 0.415 | 0.424 | 0.330 | 0.352 | 0.364 |
> |                                  |  Influence selection | 0.400 | 0.400 | 0.405 | 0.330 | 0.336 | 0.347 |
>
> The results in the table demonstrate the effectiveness of channel pruning based on the channel-wise influence function, highlighting that PatchTST and iTransformer exhibit comparable utilization of channel information on the ETTh1 and ETTm1 datasets.
>
> **Q3:** The time complexity of the proposed method is not analyzed in this paper.
>
> **A3:** We have added an experiment measuring the time required for detection at each time point to demonstrate the complexity of our approach, as shown in the table below:
>
>
> | Dataset | GCN_lstm+ours | iTransformer+ours |
> |---------|----------|--------------|
> | SWAT    |  1.4ms   |  1.5ms       |
> | WADI    |  6.4ms   |  6.5ms       |
>
> The results in the table indicate that our detection speed is at the millisecond level, which is acceptable for real-world scenarios.
>
> **Q4:** The code of this paper is not provided.
>
> **A4:** Since ICLR does not mandate code submission, we have not uploaded it at this time. However, if the paper is accepted, we commit to making our code publicly available.

---

> ### Comment · Reviewer_eNTa · 2024-11-24
>
> Thanks for the author to answer my question. My previous concerns are partially solved, and I would like to increase my score to positive.

---

> ### Author Response · Authors · 2024-11-24
>
> Thank you for your response. If you have any new questions, we would like to have a further discussion with you.

---

### Official Review · Reviewer_pGX7 · 2024-11-05

**Soundness:** 2
**Presentation:** 2
**Contribution:** 2
**Rating:** 5
**Confidence:** 5

**Summary:**

This paper introduces the Channel-wise Influence Function, a novel method tailored for multivariate time series (MTS) data to enhance model interpretability and performance by assessing the impact of individual channels. While MTS data are pivotal in domains like healthcare, traffic forecasting, and finance, traditional deep learning approaches have primarily focused on architectural improvements rather than understanding the unique contributions of each channel. Existing influence functions, effective in areas with independent data, fall short for MTS due to their inability to differentiate channel-specific effects. The proposed Channel-wise Influence Function addresses this by using a first-order gradient approximation to evaluate each channel’s contribution, proving especially useful in tasks like anomaly detection and forecasting. Extensive experiments show that this new approach outperforms traditional influence functions on real-world datasets, offering a more effective, interpretable tool for MTS analysis.

**Strengths:**

Innovation: The paper introduces the Channel-wise Influence Function, a novel method for analyzing the influence relationships between different channels in multi-channel time series (MTS) data. This approach is relatively rare in existing research and provides a new perspective for understanding and optimizing multi-channel data.
Fine-grained Dependency Analysis: Traditional influence function methods typically calculate only the overall influence, while the channel-wise influence function enables detailed quantification of each channel's influence on other channels. This fine-grained analysis is valuable in prediction and anomaly detection tasks for multi-channel data.
Broad Application Scenarios: The proposed method holds potential for various tasks, such as anomaly detection, channel pruning, and feature selection. The channel-wise influence function can help identify key channels, simplify models, and improve prediction accuracy, making it highly practical in real-world applications.

**Weaknesses:**

Lack of Clarity in Presentation: Figure 1, intended as an overview of the framework, does not clearly correspond with the main text, leaving several critical points unexplained. For instance, the calculation of the "Score" in the figure is not detailed, nor is its derivation clearly defined in the "CHANNEL-WISE INFLUENCE FUNCTION" section. Additionally, the term "well-trained model" lacks a concrete description of what type of model is being referred to. The paper also mentions that the "Channel-wise Influence" can serve as an explainable method to assess the channel-modeling capabilities of different approaches; however, it lacks detailed explanations and specific case studies to illustrate this claim.
Limited Datasets, Leading to Less Convincing Results: The experiments are conducted on a limited number of datasets, which reduces the generalizability and representativeness of the results. For example, in the time series forecasting task, only the electricity, solar-energy, and traffic datasets were used, without evaluating common benchmarks like ETTh, ETTm, and Exchange. Additionally, the forecast length was fixed at 96, which may restrict the credibility of the results and the method's broader applicability.
Insufficient Comparison with Baseline Models: The paper lacks comparison with enough baseline models, especially current mainstream state-of-the-art (SOTA) methods. This limitation makes it difficult to fully assess the proposed method's effectiveness relative to existing approaches, thus limiting the demonstration of its advantages. For instance, in time series forecasting, only PatchTST and iTransformer were used for comparison, while other competitive models like GPHT, SimMTM, TSMixer, TimesNet, and DLinear were omitted. Additionally, while the authors designed a CHANNEL PRUNING experiment, it would also be valuable to see how the Channel-wise Influence method performs in a standard time series forecasting setup for a more comprehensive evaluation.

**Questions:**

1. The framework figure (Figure 1) does not clearly correspond with the main text, particularly in areas like the calculation and derivation of "Score" and the definition of "well-trained model." Could you provide additional explanations or examples to clarify these components and better illustrate the core concept of the channel-wise influence function?
2. You mentioned that "it can serve as an explainable method to reflect the channel-modeling ability of different approaches." (6nd paragraph, line 293) Could you provide more specific explanations or examples, perhaps through case studies or illustrative examples, to demonstrate how the channel-wise influence function explains or evaluates different models' abilities to capture channel dependencies?
3. The current experiment includes only a few baseline comparisons, especially missing mainstream SOTA models like GPHT, SimMTM, TSMixer, TimesNet, and DLinear in time series forecasting. Do you plan to add comparisons with these models in future work to better demonstrate your method's competitiveness?
4. Given the limited datasets used in the experiments (electricity, solar-energy, and traffic) and the fixed forecast length of 96, the generalizability of the results might be limited. Would testing on additional datasets, such as ETTh, ETTm, and Exchange, with varied forecast lengths, help further validate the method's applicability?

---

> ### Author Response · Authors · 2024-11-17
> **Explanation of the contributions of the paper:**
>
> We appreciate your time to provide valuable comments and suggestions to improve our paper substantially. Before we begin addressing your questions, we would like to first clarify the primary contributions of our paper. Influence functions have demonstrated significant performance and value across various fields[4]-[10], with notable applications in outlier detection[7][8][9][10] and data pruning[4][5]. However, there is no preceding research on the influence functions of multivariate time series to shed light on the effects of modifying the channel of multivariate time series. To fill this gap, we propose a channel-wise influence function, which is the first data-centric method that can estimate the influence of different channels in multivariate time series. We conducted extensive experiments and found that the original influence function performed poorly in anomaly detection and could not facilitate channel pruning, underscoring the superiority and necessity of our approach. Additionally, we can further analyze the information learned by the model through the channel-wise influence matrix. For example, as mentioned in Section 5.2.1 of our paper, comparing the number of core channel subsets across different models reveals each model's capacity for capturing channel dependencies. A larger core subset indicates that the model has captured more effective information, which also highlights the interpretability of our approach.
>
> **Regarding the selection of tasks to verify our method:** Anomaly detection has long been a critical issue in multivariate time series analysis, with relevant studies including[1][2]. Data pruning is equally important as it raises the question of whether we can train a high-performing model with less data than predicted by scaling laws[3], with related work found in studies such as [3][4][5]. However, data pruning has not yet been extensively studied within the context of time series. By analyzing the characteristics of multivariate time series, we identified redundancy between channels and developed a channel pruning method based on our proposed channel-wise influence function, which outperforms traditional data pruning approaches. In addition, this method also supports the interpretability analysis of the model's ability to model multivariate time series.
>
> [1]Position paper: Quo vadis, unsupervised time series anomaly detection? ICML 2024
>
> [2]Anomaly transformer: Time series anomaly detection with association discrepancy. ICLR 2022
>
> [3]Beyond neural scaling laws: beating power law scaling via data pruning Neurips 2022
>
> [4]Data Pruning via Moving-one-Sample-out Neurips 2023
>
> [5]Dataset Pruning: Reducing Training Data by Examining Generalization Influence ICLR 2023
>
> [6]Self-influence guided data reweighting for language model pre-training. ACL 2023
>
> [7]Detecting adversarial samples using influence functions and nearest neighbors. CVPR 2020
>
> [8]Estimating training data influence by tracing gradient descent. Neurips 2020
>
> [9]Understanding Black-box Predictions via Influence Functions. ICML 2017
>
> [10]Resolving Training Biases via Influence-based Data Relabeling. ICLR 2022

---

> ### Author Response · Authors · 2024-11-17
> **Response to weaknesses:**
>
> **Q1:** Lack of Clarity in Presentation: Figure 1, intended as an overview of the framework, does not clearly correspond with the main text, leaving several critical points unexplained. For instance, the calculation of the "Score" in the figure is not detailed, nor is its derivation clearly defined in the "CHANNEL-WISE INFLUENCE FUNCTION" section.
>
> **A1：** We have revised the paper according to your suggestions and re-uploaded it. The modified sections are highlighted in blue for your reference.
>
> **Q2:** Additionally, the term "well-trained model" lacks a concrete description of what type of model is being referred to.
>
> **A2：** We have revised the paper according to your suggestions and re-uploaded it. The modified sections are highlighted in blue for your reference.
>
> **Q3:** The paper also mentions that the "Channel-wise Influence" can serve as an explainable method to assess the channel-modeling capabilities of different approaches; however, it lacks detailed explanations and specific case studies to illustrate this claim.
>
> **A3:** In our paper, we conducted interpretability analysis in *Remark 3.2 of Section 3*, as well as in the *Result Analysis* and *Outlook* sections of 5.2.1. Specifically, the size of the core subset (the red-marked proportions in Table 5 indicate the size of the core subset) reflects the model's ability to leverage information across different channels. A larger core subset indicates that the model can effectively distinguish and capture dependency between channels. This implies that the model is able to fully utilize information from various channels to enhance its predictive performance. The more information utilized, the stronger the model's capability to capture channel dependency. Therefore, the size of the core subset provides an explanation of the model's ability to model channel dependency.
>
> **Q4:** Limited Datasets, Leading to Less Convincing Results: The experiments are conducted on a limited number of datasets, which  reduces the generalizability and representativeness of the results. For example, in the time series forecasting task, only the electricity, solar-energy, and traffic datasets were used, without evaluating common benchmarks like ETTh, ETTm, and Exchange.
>
> **A4：** We proposed a novel influence function and validated its effectiveness and superiority through diverse experiments and datasets. In the context of time series forecasting, one of the objectives of our channel pruning approach is to accelerate training. Therefore, we selected datasets with relatively high computational resource demands for our experiments. To better address your suggestions, we have added experiments on the ETTh1 and ETTm1 datasets, as shown in the table below. Overall, we believe our use of a relatively large number of datasets in time-series research highlights the generalizability and effectiveness of our approach.
>
> Since the original number of channels in ETTh1 and ETTm1 is only 7, the horizontal axis in the table directly represents the number of retained channels.
>
> |              Dataset             |                      | ETTh1 |       |       | ETTm1 |       |       |
> |:--------------------------------:|:--------------------:|:-----:|:-----:|:-----:|:-----:|:-----:|:-----:|
> | number of channels retained |                      |   7   |   3   |   2   |   7   |   3   |   2   |
> |           iTransformer           | Continuous selection | 0.396 | 0.502 | 0.573 | 0.332 | 0.756 | 0.826 |
> |                                  |   Random selection   | 0.396 | 0.428 | 0.434 | 0.332 | 0.362 | 0.372 |
> |                                  |  Influence selection | 0.396 | 0.403 | 0.420 | 0.332 | 0.333 | 0.355 |
> |             PatchTST             | Continuous selection | 0.400 | 0.460 | 0.491 | 0.330 | 0.539 | 0.687 |
> |                                  |   Random selection   | 0.400 | 0.415 | 0.424 | 0.330 | 0.352 | 0.364 |
> |                                  |  Influence selection | 0.400 | 0.400 | 0.405 | 0.330 | 0.336 | 0.347 |
>
> The results in the table demonstrate the effectiveness of channel pruning based on the channel-wise influence function, highlighting that PatchTST and iTransformer exhibit comparable utilization of channel information on the ETTh1 and ETTm1 datasets.

---

> ### Author Response · Authors · 2024-11-17
> **Response to weaknesses:**
>
> **Q5:** Additionally, the forecast length was fixed at 96, which may restrict the credibility of the results and the method's broader applicability.
>
> **A5：** Thank you for your suggestion. We have added experimental results for the prediction length of 192. The detailed results are as follows:
> |              Dataset             |                      |  ECL  |       |       |       |         |       | Solar |       |       |         |         |       | Traffic |       |       |       |         |       |
> |:--------------------------------:|:--------------------:|:-----:|:-----:|:-----:|:-----:|:-------:|:-----:|:-----:|:-----:|:-----:|:-------:|:-------:|:-----:|:-------:|:-----:|:-----:|:-----:|:-------:|:-----:|
> | Proportion of variables retained |                      |   5%  |  10%  |  15%  |  20%  | **50%** |  100% |   5%  |  10%  |  15%  |   20%   | **50%** |  100% |    5%   |  10%  |  15%  |  20%  | **30%** |  100% |
> |           iTransformer           | Continuous selection | 0.212 | 0.193 | 0.189 | 0.186 |  0.182  | 0.164 | 0.270 | 0.260 | 0.256 |  0.251  |  0.249  | 0.240 |  0.486  | 0.456 | 0.427 | 0.426 |  0.425  | 0.413 |
> |                                  |   Random selection   | 0.203 | 0.189 | 0.183 | 0.179 |  0.172  | 0.164 | 0.266 | 0.258 | 0.260 |  0.249  |  0.248  | 0.240 |  0.476  | 0.436 | 0.425 | 0.421 |  0.420  | 0.413 |
> |                                  |  Influence selection | 0.191 | 0.181 | 0.173 | 0.171 |  0.165  | 0.164 | 0.259 | 0.256 | 0.254 |  0.244  |  0.242  | 0.240 |  0.460  | 0.430 | 0.422 | 0.416 |  0.413  | 0.413 |
> | Proportion of variables retained |                      |   5%  |  10%  |  15%  |  20%  | **40%** |  100% |   5%  |  10%  |  15%  | **20%** |   50%   |  100% |    5%   |  10%  |  15%  |  20%  | **20%** |  100% |
> |             PatchTST             | Continuous selection | 0.272 | 0.216 | 0.201 | 0.200 |  0.199  | 0.186 | 0.282 | 0.270 | 0.265 |  0.264  |  0.260  | 0.260 |  0.501  | 0.488 | 0.480 | 0.479 |  0.479  | 0.465 |
> |                                  |   Random selection   | 0.210 | 0.206 | 0.198 | 0.194 |  0.191  | 0.186 | 0.274 | 0.270 | 0.266 |  0.263  |  0.260  | 0.260 |  0.496  | 0.485 | 0.480 | 0.474 |  0.474  | 0.465 |
> |                                  |  Influence selection | 0.200 | 0.197 | 0.195 | 0.190 |  0.186  | 0.186 | 0.267 | 0.264 | 0.262 |  0.260  |  0.260  | 0.260 |  0.485  | 0.475 | 0.470 | 0.465 |  0.465  | 0.465 |
>
> From the results shown in the table, it can be observed that channel-pruning based on channel-wise influence is more effective. Additionally, iTransformer still exhibits a larger core subset, demonstrating its superior ability to model channel dependency.

---

> ### Author Response · Authors · 2024-11-17
> **Response to weaknesses:**
>
> **Q6:** Insufficient Comparison with Baseline Models: The paper lacks comparison with enough baseline models, especially current mainstream state-of-the-art (SOTA) methods. This limitation makes it difficult to fully assess the proposed method's effectiveness relative to existing approaches, thus limiting the demonstration of its advantages. For instance, in time series forecasting, only PatchTST and iTransformer were used for comparison, while other competitive models like GPHT, SimMTM, TSMixer, TimesNet, and DLinear were omitted.
>
> **A6：**  Thank you for your suggestion. Considering that the primary purpose of our method is to accelerate training and better interpret the model's behavior, we did not include comparisons with a wide range of time series forecasting methods. However, given that DLinear is a highly representative approach, we have added it as a supplementary method and tested it under our original setting. The results are as follows:
> |              Dataset             |                      |  ECL  |       |       |       |       |       | Solar |       |       |       |       |       | Traffic |       |       |       |       |       |
> |:--------------------------------:|:--------------------:|:-----:|:-----:|:-----:|:-----:|:-----:|:-----:|:-----:|:-----:|:-----:|:-----:|:-----:|:-----:|:-------:|:-----:|:-----:|:-----:|:-----:|:-----:|
> | Proportion of variables retained |                      |   5%  |  10%  |  15%  |  20%  |  50%  |  100% |   5%  |  10%  |  15%  |  20%  |  50%  |  100% |    5%   |  10%  |  15%  |  20%  |  30%  |  100% |
> |              DLinear             | Continuous selection | 0.201 | 0.200 | 0.198 | 0.197 | 0.196 | 0.196 | 0.311 | 0.309 | 0.307 | 0.301 | 0.301 | 0.301 |  0.649  | 0.647 | 0.645 | 0.645 | 0.645 | 0.645 |
> |                                  |   Random selection   | 0.200 | 0.198 | 0.196 | 0.196 | 0.196 | 0.196 | 0.306 | 0.304 | 0.303 | 0.301 | 0.301 | 0.301 |  0.649  | 0.648 | 0.645 | 0.645 | 0.645 | 0.645 |
> |                                  |  Influence selection | 0.197 | 0.196 | 0.196 | 0.196 | 0.196 | 0.196 | 0.301 | 0.301 | 0.301 | 0.301 | 0.301 | 0.301 |  0.646  | 0.645 | 0.645 | 0.645 | 0.645 | 0.645 |
>
> The experimental results in the table show that the core channel subset of DLinear is less than 5%, which highlights the limited ability of simple linear models to utilize information from different channels effectively.
>
> **Q7:** Additionally, while the authors designed a CHANNEL PRUNING experiment, it would also be valuable to see how the Channel-wise Influence method performs in a standard time series forecasting setup for a more comprehensive evaluation.
>
> **A7：** We are not entirely sure we understand your question on this point. As described in Section 4.2 of our paper, the motivation behind channel pruning is to accelerate training and enable interpretability analysis, rather than to propose a method for improving time series forecasting performance. Therefore, we did not conduct experiments under the standard time series settings. If we have misunderstood your concerns, we would greatly appreciate your clarification.

---

> > ### Author Response · Authors · 2024-11-17
> > **Response to Questions:**
> >
> > **Q1:** The framework figure (Figure 1) does not clearly correspond with the main text, particularly in areas like the calculation and derivation of "Score" and the definition of "well-trained model." Could you provide additional explanations or examples to clarify these components and better illustrate the core concept of the channel-wise influence function?
> >
> > **A1：** We have revised the paper based on your suggestions and marked the changes in blue. The specific explanations are as follows:
> >
> > **Score** refers to the influence score calculated through the channel-wise self-influence function, specifically the diagonal elements of the channel-wise influence matrix.
> > **Well-trained model** denotes a model that has been trained on any given task.
> > The core of the channel-wise influence function lies in its ability to measure the influence of different channels between any two samples. This is distinct from the original influence function, which can only evaluate the influence between two samples as a whole.
> >
> > With the channel-wise influence function, we can determine the influence between different channels within a single sample. This, to some extent, reflects the dependency between different channels, as discussed in Remark 3.2 of the paper.
> >
> > **To illustrate with a specific example**: suppose we have a multivariate time series with 100 channels. By calculating the channel-wise influence function for this sample, we can obtain the influence between any two channels. According to the definition of the influence function, it represents how much training on channel a helps reduce the test loss of channel b. This means that channels with high influence are highly similar, and selecting only a representative subset of channels can enable the model to make predictions across all channels. For instance, if we train on just 10 representative channels, the model can generalize to the remaining 90 channels because these 10 channels exhibit high influence with the others.
> >
> > **Q2:** You mentioned that "it can serve as an explainable method to reflect the channel-modeling ability of different approaches." (6nd paragraph, line 293) Could you provide more specific explanations or examples, perhaps through case studies or illustrative examples, to demonstrate how the channel-wise influence function explains or evaluates different models' abilities to capture channel dependencies?
> >
> > **A2：** As addressed in Response to Weaknesses under A3, we explain the quality of a model's channel modeling capability by analyzing the size of its core channel subset (the red-marked proportions in Table 5 indicate the size of the core channel subset). Specific experimental examples are shown in Section 5.2.1, and detailed experimental analyses are provided in the Result Analysis and Outlook sections.
> >
> > **Q3:** The current experiment includes only a few baseline comparisons, especially missing mainstream SOTA models like GPHT, SimMTM, TSMixer, TimesNet, and DLinear in time series forecasting. Do you plan to add comparisons with these models in future work to better demonstrate your method's competitiveness?
> >
> > **A3：** We have added new experiments as per your request. Please refer to *Response to Weaknesses* sections *A6* for the details.
> >
> > **Q4:** Given the limited datasets used in the experiments (electricity, solar-energy, and traffic) and the fixed forecast length of 96, the generalizability of the results might be limited. Would testing on additional datasets, such as ETTh, ETTm, and Exchange, with varied forecast lengths, help further validate the method's applicability?
> >
> > **A4：** We have added new experiments as per your request. Please refer to *Response to Weaknesses* sections *A4 and A5* for the details.

---

> > > ### Author Response · Authors · 2024-11-24
> > > **Gentle reminder**
> > >
> > > Dear Reviewer pGX7
> > >
> > > We sincerely appreciate your time and effort in reviewing our manuscript and offering valuable suggestions. As the discussion time is coming to an end, we would like to confirm whether our responses have effectively addressed your concerns. We provided detailed responses to your concerns a few days ago, and we hope they have adequately addressed your issues. If you require further clarification or have any additional concerns, please do not hesitate to contact us. We are more than willing to continue our communication with you.
> > >
> > > Best regards,

---

> > > > ### Comment · Reviewer_pGX7 · 2024-11-25
> > > > **Response to authors**
> > > >
> > > > We appreciate the authors' thoughtful and detailed response.

---

> ### Author Response · Authors · 2024-11-25
> **Response to Reviewer**
>
> Thanks for the responses. We sincerely and respectfully hope that you might consider reevaluating our work in light of the responses we have provided, which aim to address your main concerns.  Your insights and contributions will be a nice addition to our community.
>
> We greatly appreciate your time and effort in advance.  If you have other concerns, we are willing to further address them.

---

> > ### Comment · Reviewer_pGX7 · 2024-12-03
> > **One more question?**
> >
> > Could you further clarify under what conditions or scenarios, such as varying numbers of channels, the strengths and limitations of your method become apparent?

---

> > > ### Author Response · Authors · 2024-12-03
> > > **Response to New question**
> > >
> > > Our method demonstrates significant advantages when it comes to accurately identifying the influence of a specific channel within an MTS. Specifically, as highlighted in our paper, this includes anomaly detection tasks where pinpointing the exact anomalous channel is crucial, as well as channel pruning tasks aimed at identifying the most representative channels to accelerate model training—an advantage that becomes more pronounced as the number of channels increases. Furthermore, our method can be applied to tasks like data attribution, where understanding which channel significantly influenced a model's decision is essential.
> > >
> > > Regarding the limitations of our approach, one challenge lies in deployment on certain edge devices, as some may not support gradient computation, leading to practical restrictions. Additionally, while the computation time increases with the number of channels, this growth is approximately linear, as detailed in Appendix C.4.

---

> > > > ### Author Response · Authors · 2024-12-03
> > > > **Gentle Reminder**
> > > >
> > > > Dear Reviewer pGX7,
> > > >
> > > > We sincerely appreciate your time and effort in reviewing our manuscript and offering valuable suggestions. As the discussion phase is drawing to an end, we sincerely hope to know whether we have addressed all of your concerns. If you have any further questions, we would be glad to engage in further discussion.
> > > >
> > > > Best Regards

---

### Note · Authors · 2025-01-23

I have read and agree with the venue's withdrawal policy on behalf of myself and my co-authors.